# Metastatic small cell lung cancer arises from TP53/RB1-deficient and MYC overproduction hESC-derived PNECs

Huanhuan Joyce Chen[1,2,3]*[†], Eric E Gardner[1][†], Yajas Shah[4], Kui Zhang[2,3], Abhimanyu Thakur[2,3], Chen Zhang[5,6], Olivier Elemento[4,6], Harold Varmus[1]*

[1]Meyer Cancer Center, Weill Cornell Medicine, New York, United States; [2]The Pritzker School of Molecular Engineering, The University of Chicago, Chicago, United States; [3]The Ben May Department for Cancer Research, The University of Chicago, Chicago, United States; [4]Caryl and Israel Englander Institute for Precision Medicine, Weill Cornell Medicine, New York, United States; [5]Department of Pathology and Laboratory Medicine, Weill Cornell Medicine, New York, United States; [6]Institute for Computational Biomedicine, Weill Cornell Medicine, New York, United States

*For correspondence:
joycechen@uchicago.edu (HJC);
varmus@med.cornell.edu (HV)

[†]These authors contributed equally to this work

Competing interest: The authors declare that no competing interests exist.

## eLife Assessment

Given a great need for novel human model systems to study small cell lung cancer (SCLC), the authors describe an **important** pre-clinical model with broad potential for the study of how genetic perturbations or drug treatments alter SCLC tumor growth, metastasis, and response to therapy. For the major finding, the authors provide **convincing** evidence that RB/TP53 suppression coupled with MYC overexpression in an ES cell-derived model system results in aggressive and metastatic SCLC. However, the impact of the work would have been increased with the inclusion of a broader set of genetic perturbations, such as over-expression of MYCL, to better model major SCLC phenotypes. The new model described will be of significant interest to researchers studying lung cancer.

**Abstract** We previously described our initial efforts to develop a model for small cell lung cancer (SCLC) derived from human embryonic stem cells (hESCs) that were differentiated to form pulmonary neuroendocrine cells (PNECs), a putative cell of origin for neuroendocrine-positive SCLC. Although reduced expression of the tumor suppressor genes TP53 and RB1 allowed the induced PNECs to form subcutaneous growths in immune-deficient mice, the tumors did not display the aggressive characteristics of SCLC seen in human patients. Here, we report that the additional, doxycycline-regulated expression of a transgene encoding wild-type or mutant MYC protein promotes rapid growth, invasion, and metastasis of these hESC-derived cells after injection into the renal capsule. Similar to others, we find that the addition of MYC encourages the formation of the SCLC-N subtype, marked by high levels of NEUROD1 RNA. Using paired primary and metastatic samples for RNA-sequencing, we observe that the subtype of SCLC does not change upon metastatic spread and that production of NEUROD1 is maintained. We also describe histological features of these malignant, SCLC-like tumors derived from hESCs and discuss potential uses of this model in efforts to control and better understand this recalcitrant neoplasm.

## Introduction

From the earliest days of cancer research, attempts to understand the mechanisms of carcinogenesis have depended on experimental models to circumvent the ethical obstacles to performing experiments in human subjects. Such model systems have ranged from normal and cancerous human cells grown in animals or in cell culture to the generation or manipulation of cancers arising from normal cells in whole animals. The success of such approaches is often measured by the similarities of model systems to the observed traits of cancers that have arisen naturally in human subjects. But the utility of certain model systems can also be appraised by evaluation of the information that emerges from them, even when the models lack a clear relationship to any specific type of human cancer. The transformation of chicken embryo fibroblasts by Rous sarcoma virus or the induction of leukemias with various murine leukemia viruses exemplifies models that lack verisimilitude but nevertheless have revealed major features of carcinogenesis (*Varmus and Weinberg, 1993*).

A common complaint about many models of cancer is that the transformation of a normal cell to a malignant cell occurs in nonhuman cells. Efforts to transform human cells are often limited by the difficulties of growing human cells in culture and uncertainty about the developmental stage of a given cell lineage at which transformation generally occurs. Techniques for generating, propagating, and differentiating human stem cells – either tissue-specific stem cells or pluripotent stem cells – have raised the possibility of studying carcinogenesis in a wide variety of human cell lineages, as initially demonstrated for gliomas derived from induced human pluripotent stem cells (*Funato et al., 2014*).

We have recently attempted to study the mechanisms by which small cell lung cancer (SCLC) arises in human lung cells by efficiently differentiating human embryonic stem cells (hESCs) (*Huang et al., 2015*; *Huang et al., 2014*) into mature lung epithelial cells, including the likely precursor of SCLC (pulmonary neuroendocrine cells [PNECs]) and then simulating loss of function of the tumor suppressor genes *TP53* and *RB1*, tumor suppressors commonly lost in human SCLC (*Chen et al., 2019*). PNEC-like cells that have lost the function of these two tumor suppressor genes, by expression of short hairpin RNAs (shRNAs) specific for *RB1* and *TP53*, can form tumors subcutaneously in immunodeficient mice; however, these tumors grow slowly and do not invade or metastasize, standing in contrast to what is known about the clinical presentation and behavior of SCLC (*Budczies et al., 2015*). Thus, the original hESC-derived SCLC models we generated appeared to be benign growths that do not resemble SCLCs functionally, limiting their utility as models for aggressive human SCLC.

Our earlier findings are consistent with the idea that full transformation of PNECs to form malignant SCLC requires additional changes beyond restricted expression of the tumor suppressor genes *RB1* and *TP53*, such as heightened expression of a proto-oncogene in the *MYC* family (*Ireland et al., 2020*; *Mollaoglu et al., 2017*). In an attempt to show that we can generate more aggressive, SCLC-like tumors from PNECs derived from hESCs, we have added efficiently transcribed MYC oncogenes to cells in which expression of TP53 and RB1 is inhibited by shRNAs. When these cells were then injected into renal capsules of immunodeficient mice, some produced rapidly expanding tumors that invaded locally and sometimes formed metastatic growths, mostly in the liver.

In this report, we describe morphological and transcriptional features of these tumors and discuss how the addition of putative drivers of SCLCs can be used to study the mechanism of carcinogenesis and to test candidate therapies for this 'recalcitrant' cancer.

## Results

### Generation of hESC-derived PNEC-like cells that express transgenes encoding MYC

We used lentivirus vectors encoding doxycycline (DOX)-inducible mRNAs for wild-type (WT) and mutant (T58A) human MYC (see Materials and methods) to infect previously described hESCs from the RUES2 cell line carrying transgenes encoding DOX-inducible shRNAs complementary to *TP53* and *RB1*. For simplicity of presentation, we refer to the latter RUES2 cells as RP cells, and the RP cells containing DOX-inducible WT or T58A mutant MYC as RPM and RPM (T58A) cells throughout the text and figures of this paper.

After the growth of infected cells and differentiation into lung cell progenitors (LPs) and mature lungcells (LCs) (*Figure 1A*), we induced PNECs by inhibition of NOTCH signaling with DAPT, as previously reported (*Chen et al., 2019*). The ability of the RPM and RPM (T58A) cells to produce MYC

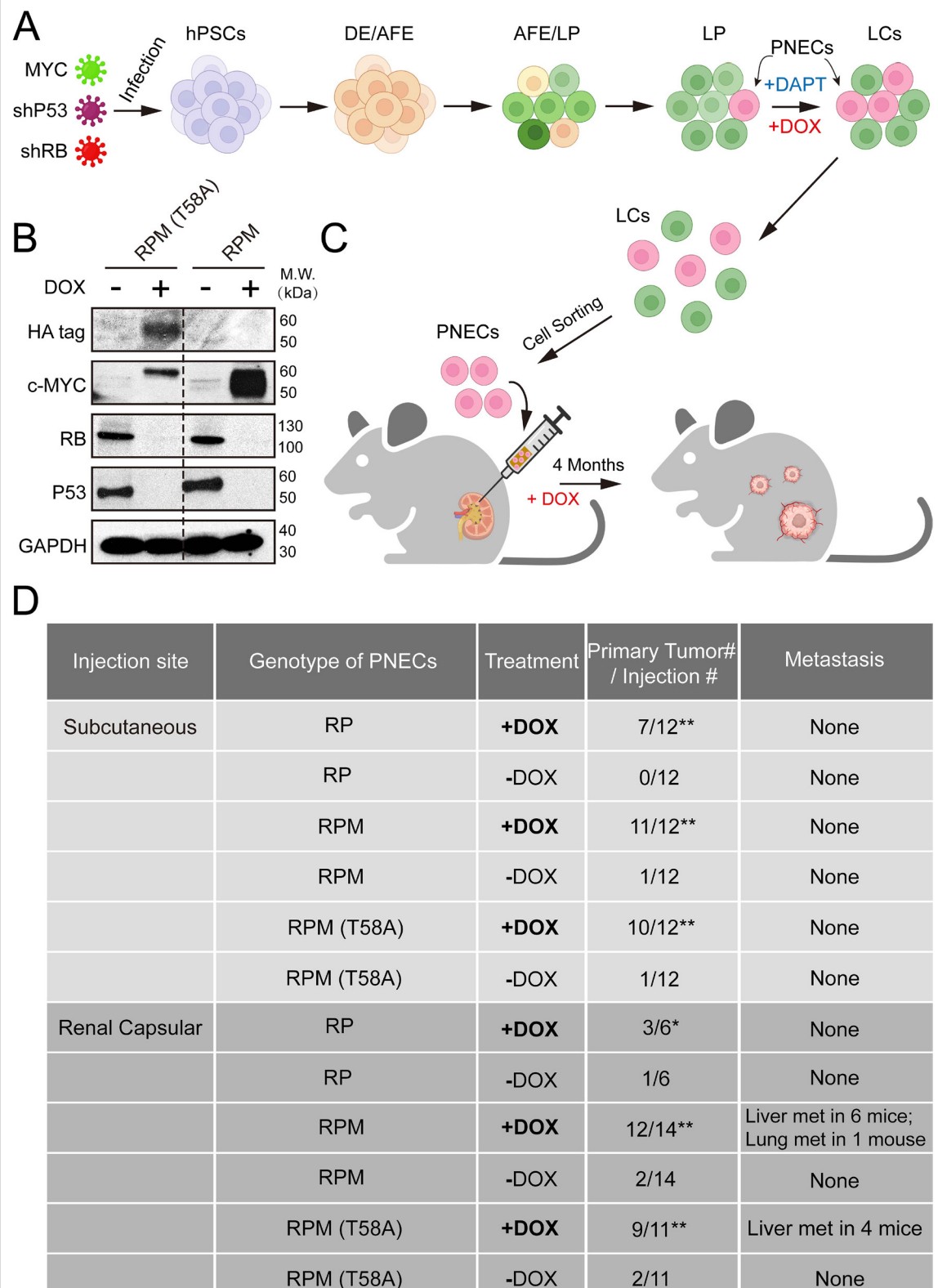

**Figure 1.** Generation and characterization of human embryonic stem cell (hESC)-derived lung cells and formation of xenografts with RPM cells. (**A**) Schematic of the protocol used to generate pulmonary neuroendocrine cells (PNECs) by stepwise differentiation of human pluripotent stem cells (hPSCs) to form: definitive endoderm (DE), day 3; anterior foregut endoderm (AFE), day 6; and lung progenitor cells (LPs) days 15–25. LPs were further differentiated into the types of lung cells (LCs) found in mature human lung parenchyma and airway epithelium, days 25–55. DAPT (10 μM) encourages

*Figure 1 continued on next page*

*Figure 1 continued*

the formation of PNECs, and addition of doxycycline (1 µM; DOX) induces expression of shRNAs against RB1 and TP53 mRNAs, as well as expression of MYC or MYC (T58A), as described in the text. (**B**) Western blot of extracts of RUES2 LCs at day 25 of differentiation protocol treated with DOX (1 µM for 72 hr); cells unexposed to DOX served as negative expression controls. Apparent differences in MYC protein levels may be attributable to the HA-tagged version of MYC (T58A), which migrates slightly slower than wild-type MYC protein. (**C**) Schematic representation of tumorigenesis experiments comparing injection sites (renal capsule or subcutaneous), DOX treatment (+/-DOX diet), and genotypes (see Materials and methods for additional details). Total numbers of animals are six to seven per experimental arm with two injection sites per mouse (right and left flank). Renal capsule injections were performed on a single kidney. Transgenic lines of RUES2 hESCs were differentiated and grown in DAPT (10 µM) from days 25 to 55. At day 55, PNECs were separated from other LCs by sorting for PE+ CGRP-expressing cells (see Materials and methods). PNECs were then injected either subcutaneously or into the renal capsular space in NOG mice, half of which then received DOX in their feed as described in Materials and methods. (**D**) Table summary of experiments with xenografted mice, indicating the number of animals that developed visible tumors (≥250 mm$^3$ in volume) at the site of injection or the number of visible metastases in the liver or lung. *, p<0.05; **, p<0.01 by Fisher's test to denote significant differences between mice that did not receive DOX diet. As before, abbreviations for cell lines are: RP = shRB1+shTP53; RPM = shRB1+shTP53+WT MYC; RPM (T58A)=shRB1+shTP53+MYC (T58A).

The online version of this article includes the following source data for figure 1:

**Source data 1.** Raw data of western blot bands in *Figure 1*.

**Source data 2.** Raw data of *Figure 1*.

protein was documented by western blotting of cell extracts prepared 72 hr after addition of DOX (*Figure 1B*). As expected, both cell lines also contained dramatically reduced amounts of TP53 and RB1 proteins as a result of DOX induction of shRNAs specific for these tumor suppressor mRNAs (*Figure 1B*).

## RUES2-derived PNECs that express WT or T58A MYC show enhanced oncogenicity as xenografts in the renal capsule of immunodeficient mice

To determine whether production of WT or mutant MYC proteins renders PNECs derived from RPM lines more oncogenic, we sorted CGRP-positive RPM and RPM (T58A) cells from cultures of LCs treated with DAPT and injected the cells subcutaneously or into the renal capsule of immunodeficient mice (the NOG strain: NOD/Shi-scid/IL-2Rγnull) as depicted in *Figure 1A*. The mice were then maintained on diets with or without DOX (see Materials and methods). Three to four months later, subcutaneous growths larger than 1000 mm$^3$ or renal engraftments greater than 100 mm$^3$ and grossly metastatic lesions in the liver and lungs were tabulated, as summarized in *Figure 1C and D*. As previously reported, PNECs from cultures of RP lines of RUES2 cells produced small benign lesions subcutaneously without invasion or metastasis. Similar findings were observed when these cells were injected into the renal capsule (*Figure 1D*).

## Growth of RUES2-derived RPM tumors is DOX-dependent

Thus far, our data suggest that at least some RUES2-derived, PNEC-like cells in which expression of two tumor suppressor genes, *RB1* and *TP53*, is restricted, RP cells, can be transformed into aggressive, metastatic cells by the addition of transgenes encoding WT or mutant MYC protein (hereafter referred to as 'RPM' or 'RPM (T58A)' cells). However, we had not directly compared the relative growth rates of these engineered cell types.

We therefore engrafted immunocompromised mice subcutaneously with equivalent numbers of RP, RPM, or RPM (T58A) cells and monitored tumor growth rates. We noted that the volumes of RPM and RPM (T58A)-derived tumors quickly exceeded the volumes of RP-derived tumors (*Figure 2A*). The rapid growth of RPM tumors was dependent on continuous provision of DOX in the diet. If animals on a DOX-free diet were engrafted subcutaneously with RPM or RPM (T58A) cells, they failed to produce tumors (*Figure 2B*, left). However, tumor growth was observed in some mice engrafted with such cells if the animals were placed on a DOX-containing diet up to 2 months after injection (*Figure 2B*, right). These findings suggest that at least some of the RPM cells remained viable and responsive to the oncogenic effects of depleting RB1 and TP53 and producing MYC.

The rapid growth and morphological features of RPM and RPM (T58A) tumors in these experiments were histologically consistent with SCLC. Critically, subcutaneous injection did not produce metastatic disease to the liver (*not shown*), nor was there evidence of gross metastases to the lungs in any of

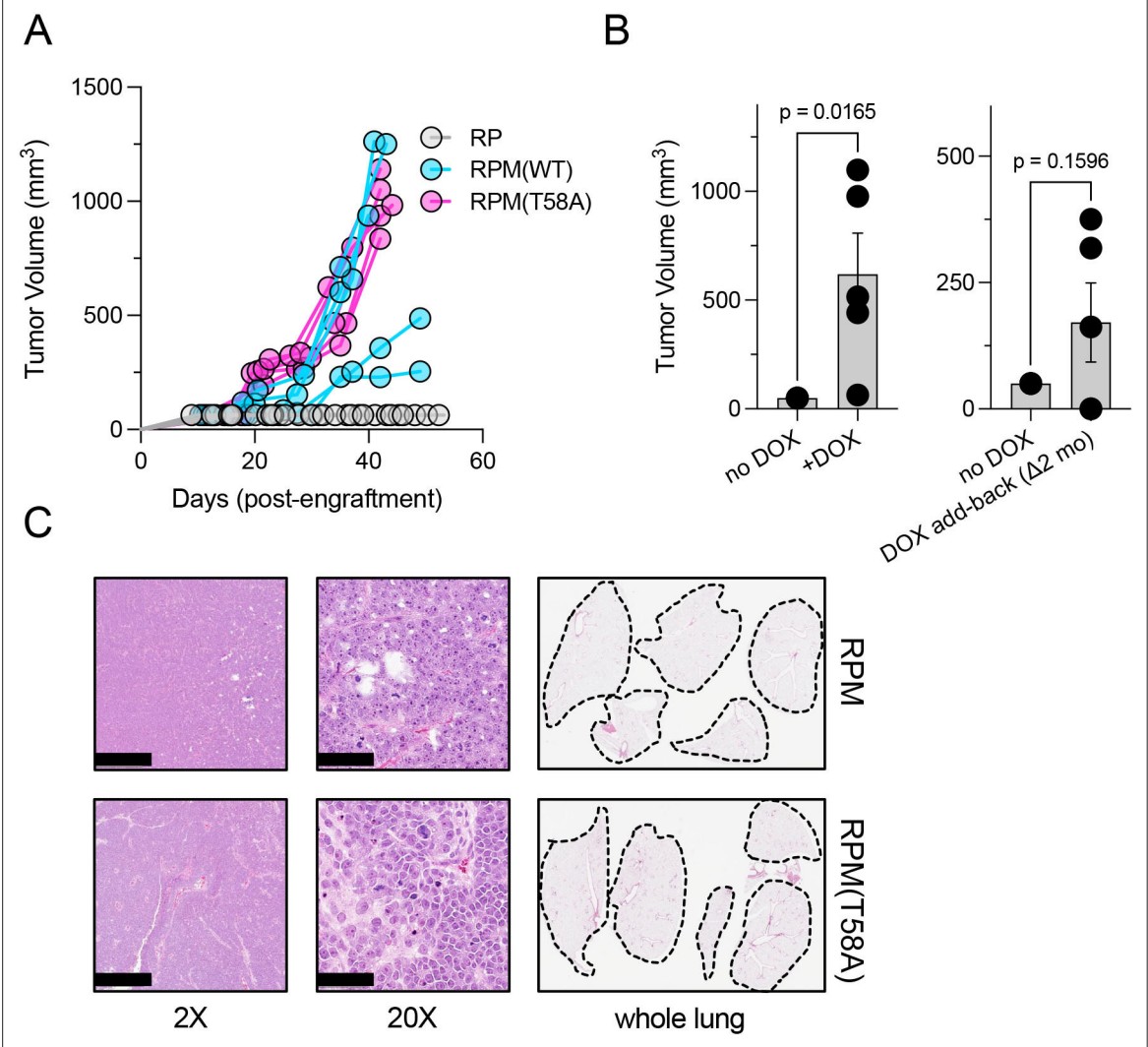

**Figure 2.** Dox-dependent growth of RUES2-derived RPM and RPM (T58A) subcutaneous tumors. (**A**) Subcutaneous engraftment of $10^6$ viable cells at ~day 50 of RUES2 lung differentiation protocol from indicated genotypes. Cells were cultured in the presence of media containing 1 μM doxycycline (DOX) and 10 μM DAPT from days 25 to 55. All immunocompromised mice were maintained on the DOX diet throughout this study; n=5/arm. (**B**) Left: subcutaneous engraftment of $10^6$ viable RPM tumor cells from the first passage (mouse-to-mouse passage) into immunocompromised mice (NSG) maintained on DOX or normal chow; n=5 animals per arm with single flank engraftments, ± standard error of the mean (SEM). **p<0.05. Right: after 1 month on normal chow, mice were placed on DOX chow (DOX 'add-back'); tumor volumes were measured 1 month later (2 months on study). Paired t-test p-values are shown for each comparison. (**C**) Representative H&E tumor histology from RPM or RPM (T58A) engraftments; 2× (scalebar 1 mm) and 20× (scalebar 100 μm). Representative whole lung sagittal sections showed no evidence of gross metastasis from subcutaneous injections inthe flank at study endpoints; n=3 analyzed per genotype. Dashed lines shown for boundaries of lung lobes.

The online version of this article includes the following source data for figure 2:

**Source data 1.** High resolution source data of *Figure 2*.

the mice examined (*Figure 2C*). These findings suggest that SCLC-like tumors composed of RUES2-derived RPM cells grow rapidly after subcutaneous implantation, but may require a different micro-environment, such as the vascular-rich, highly perfused renal capsule (*Bogden et al., 1979*; *Sobczuk et al., 2020*), providing a more hospitable environment for metastatic seeding.

## Inoculation of the renal capsule facilitates metastasis of the RUES2-derived RPM tumors

RPM tumors, as compared to RP tumors, were notable for increased vascularization, larger volume, and invasion into surrounding endothelium. RPM and RPM (T58A) PNECs injected into the renal

capsule of immunocompromised mice also displayed frequent metastasis to the liver (*Figure 3D*). In animals administered DOX, histological examinations showed that approximately half developed metastases in distant organs, including the liver or lung. One of the mice developed metastatic tumors in both liver and lung sites (*Figure 1D*). No metastases were observed in the bone, brain, or lymph nodes. Metastatic lesions were characterized as having similar proliferative character and expression of MYC, as well as comparable expression of the neuronal markers NCAM (CD56) and NEUROD1 compared to their paired primary counterparts (*Figure 3E and F*). In contrast, ASCL1 expression was absent in RPM and RPM (T58A) tumors at both primary and metastatic sites (*Figure 3F*). Meanwhile, the majority of cells in RP tumors showed positive ASCL1 expression but lacked NEUROD1 expression (*Figure 3G*). These findings suggest that MYC expression may facilitate the transition of SCLC tumors from the classical ASCL1-driven subtype to the NEUROD1 variant. Thus, both the site of implantation of cells and the addition of a driver oncogene were required to produce vigorous growth and metastatic spread of SCLC-like tumors, and the metastatic lesions retained histological and immunohistochemical features of the primary tumors.

## RUES2-derived RPM tumors most closely resemble the NEUROD1-high (SCLC-N) subtype of SCLC

To better understand the characteristics of RUES2-derived RPM tumors, we performed single-cell RNA-seq (RNA-seq) on culture-derived ('2D') and tumor-derived RPM and RPM (T58A) viable EGFP+ cells. By integrating previously published RUES2-derived RP single-cell data (*Chen et al., 2019*), we observed three dominant cell-type groups in transcriptional space, using reference mapping, enrichment analysis, and manual curation to identify cell lineage likelihood: a basal-like cluster marked by expression of KRT19, a fibroblast-like cluster marked by expression of *COL1A1*, and the most prominent cluster of cells appearing to have features of neuroendocrine differentiation, marked by expression of the neuroendocrine lineage defining transcription factor *ASCL1* (*Figure 4A* and *Supplementary file 1a*). Comparing the recently described SCLC subtype markers (*Rudin et al., 2019*) across these datasets showed that cells within the neuroendocrine differentiation cluster cells expressed *ASCL1* and *NEUROD1*, whereas *POU2F3 RNA* was not detected. Levels of *YAP1* RNA were variable and were not co-expressed with either *ASCL1* or *NEUROD1* (*Figure 4B*). *YAP1* expression was mostly attributed to fibroblast- and basal-like cells that are often observed in long-term cultures of differentiated hESCs (*Figure 4B*; *Huang et al., 2014*; *Chen et al., 2017*; *Dye et al., 2015*; *Rodrigues Toste de Carvalho et al., 2021*).

Next, we asked whether the overexpression of *MYC* in RPM cells was associated with increased neuroendocrine differentiation when compared to cell lines that did not contain MYC transgenes (i.e. RP cells). One notable difference between our prior work and this study was our attempt to increase the tumor cell component of the RPM and RPM (T58A) samples by sorting cells for high EGFP expression. EGFP is expressed in *cis* with the RB1 targeting shRNA; however, EGFP is not expressed uniformly across all cells following the differentiation protocol, suggesting some cells may silence the integrated vector. We performed differential cluster abundance analysis after accounting for the fraction of cells that were EGFP+ during cell sorting. These results indicated that RPM cell lines (WT or T58A) were associated with a twofold increase in the neuroendocrine compartment when compared with RP cells (p<2.2 × 10⁻¹⁶; *Figure 4—figure supplement 1A and B*). Similarly, RPM tumors had a threefold enrichment of cells expressing neuroendocrine markers compared to RP cell lines (p<2.2 × 10⁻¹⁶). Notably, we found that RPM tumors and cells expressed MYC, not MYCN or MYCL at high levels (*Figure 4—figure supplement 1C*). Moreover, EGFP expression was strongly correlated with MYC (Pearson r=0.45, p<2.2 × 10⁻¹⁶) and not MYCN (Pearson r=0.00043, p=0.97) nor MYCL (Pearson r=–0.031, p=0.002). Together, these results suggest that the overexpression of either WT or T58A MYC yielded a larger neuroendocrine compartment.

By sub-clustering the neuroendocrine differentiation group, we could clearly identify ASCL1-high and NEUROD1-high compartments, as well as several other groups of cycling cells that did not cleanly fit into any lung lineage cell identity (*Figure 4D*, *Supplementary file 1*). We conducted differential gene expression to further characterize the NEUROD1-high (abbreviated 'NE-variant') and ASCL1-high (abbreviated 'NE') clusters observed with RPM cells (from cell lines and tumors) and found that the NE-variant cluster was associated with increased expression of genes highly expressed in variant SCLC (NEUROD1, SCG3, IGFBPL1, SSTR2, and MYC) (*Figure 4—figure supplement 1D*,

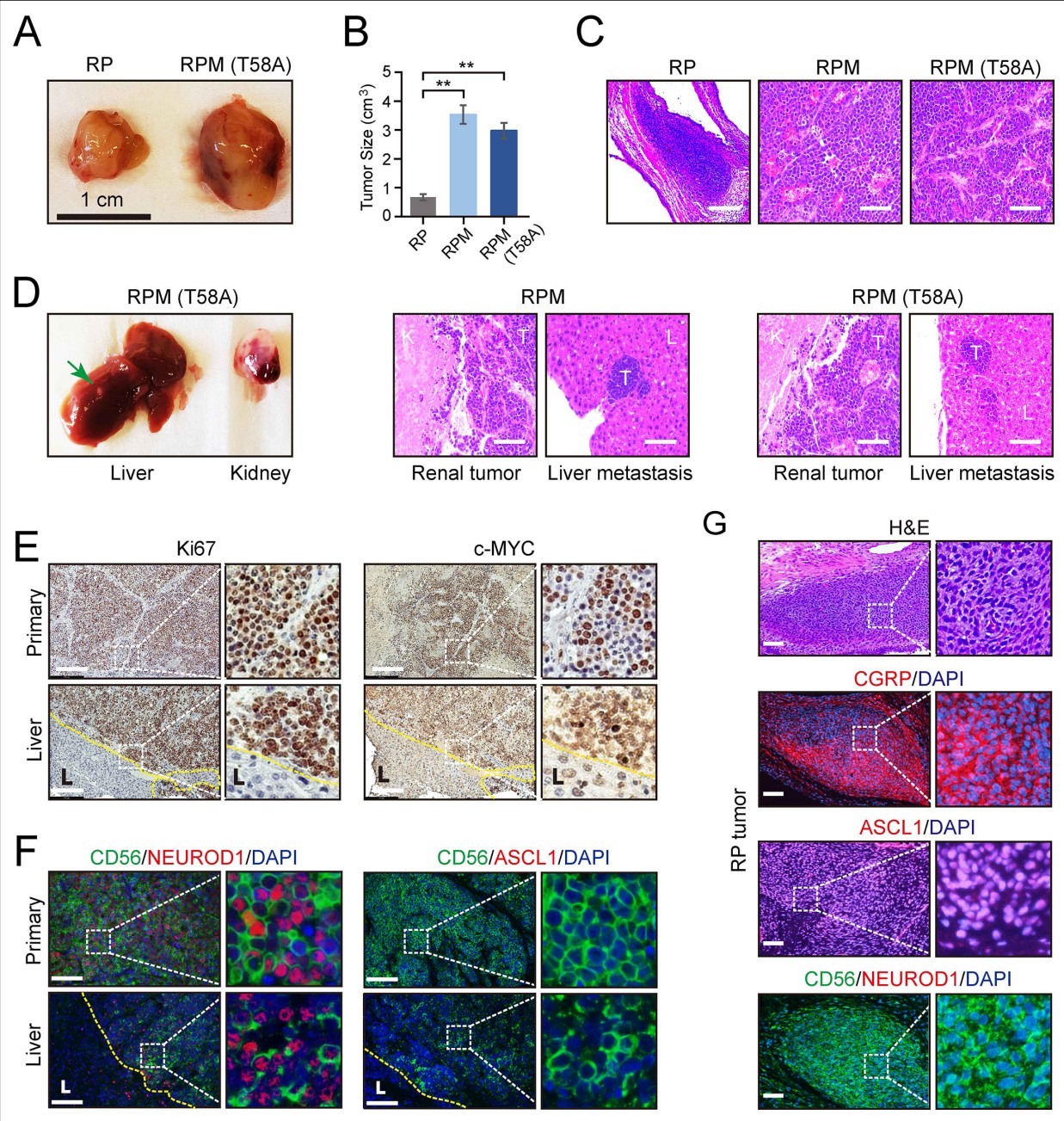

**Figure 3.** Effects of MYC on the gross and histopathological appearance of xenografts grown in immunodeficient mice fed with doxycycline (DOX) for 3–4 months. (**A**) Subcutaneous xenografts formed with human embryonic stem cell (hESC)-derived pulmonary neuroendocrine cells (PNECs) with RP, RPM, or RPM (T58A) genotypes. Photographs of representative tumors formed with cells of the indicated genotypes are shown for RP and RPM; indicated scale of 1 cm. (**B**) Quantification of the tumor sizes and paired comparisons for a secondary in vivo experiment; n=5 animals per arm with single subcutaneous engraftments; **p<0.01. (**C**) H&E staining of the indicated tumors from panel B. (**D**) Gross and histologic pathology of renal capsular xenografts and liver metastases formed with the RPM or RPM (T58A) cells. Left panels, gross appearance of representative tumors within the liver (metastasis) or kidney (primary); right panels, H&E staining of primary and metastatic tumors (**T**) formed in kidney (**K**) and liver (**L**) in the RPM (left) or RPM (T58A) model. (**E**) Immunostaining of tumor samples from D. Samples of primary peri-renal tumors and hepatic metastases in mice injected with hESC-derived PNECs programmed to reduce levels of TP53 and RB1 mRNA and to express wild-type MYC were stained with antisera for Ki67 (left) or MYC (right). (**F**) Immunofluorescence staining for neuroendocrine markers ASCL1, NEUROD1, and CD56 from sections in E of the RPM tumors. (**G**) Immunofluorescence staining for neuroendocrine markers, CGRP, ASCL1, NEUROD1, and CD56 in the RP tumors. H&E staining serves as the bright-field comparison of the indicated tumors.

The online version of this article includes the following source data for figure 3:

**Source data 1.** High resolution source data of *Figure 3*.

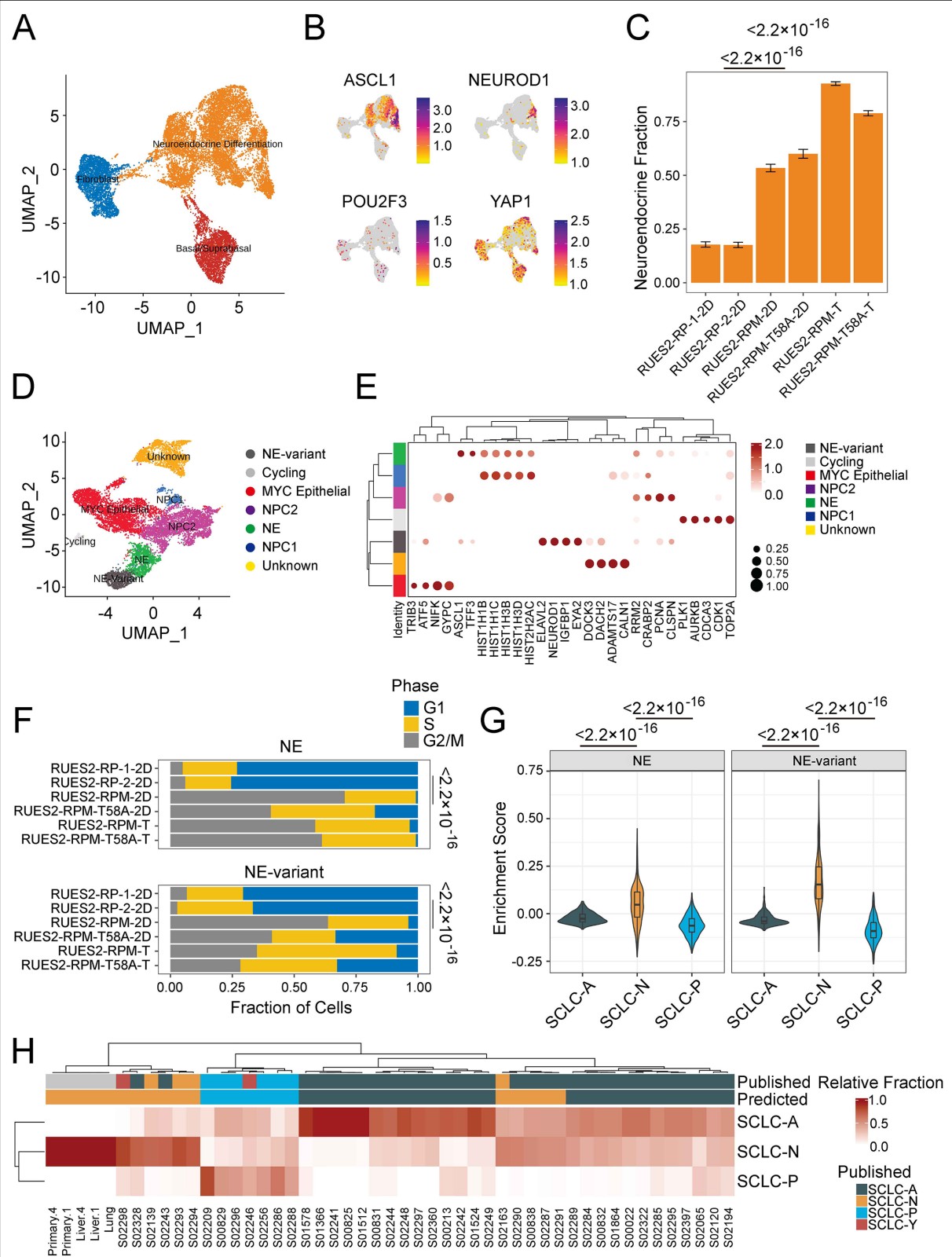

**Figure 4.** Single-cell and bulk RNA profiling of RUES2-derived RPM tumors and comparison with primary human small cell lung cancer (SCLC). (**A**) Uniform Manifold Approximation and Projection (UMAP) of cultured RP samples from *Morse et al., 2019*, and RPM samples (*this study*) with three major cellular lineages annotated by color. (**B**) Expression of SCLC subtype markers across dataset in A. (**C**) Neuroendocrine fraction of cells by sample ID. (**D**) Subclustering of the 'neuroendocrine differentiation' cluster from A. (**E**) Dot plot of differential cluster markers in the subclustering analysis from

*Figure 4 continued on next page*

*Figure 4 continued*

E. (**F**) Cell cycle evaluation of NE and NE-variant clusters indicated by color and fraction of cells. (**G**) SCLC subtype enrichment scores from *Chan et al., 2021*; cluster markers in NE and NE-variant cells from RPM tumors. (**H**) Bulk RNA-sequencing subtype estimation based on *Chan et al., 2021*, SCLC subtypes from RPM primary or metastatic tumors as compared to published primary SCLC data. Published labels were obtained from *Rudin et al., 2019*. Bulk patient RNA-sequencing data (reads per kilobase per million mapped reads [RPKM]) were compared to bulk RNA-sequencing of select RPM tumors and metastatic samples. Primary tumor and liver metastases were obtained from two pairs (animals #1 and #4) of mice engrafted with RUES2-derived RPM tumor cells into the renal capsule and grossly macro-dissected during necropsy.

The online version of this article includes the following source data and figure supplement(s) for figure 4:

**Source data 1.** High resolution source data of *Figure 4*.

**Figure supplement 1.** Gene expression and characterization of RUES2-derived tumors.

**Figure supplement 1—source data 1.** MYC family member expression correlates in RUES2-derived RPM tumors.

**Figure supplement 2.** MYC family member expression correlates in RUES2-derived RPM tumors.

**Figure supplement 2—source data 1.** MYC family member expression correlates in RUES2-derived RPM tumors.

*Supplementary file 1*). In contrast, the NE (or 'NE-classic') cluster was linked to heightened expression of a variety of genes ranging from histones (HISTH1B, HISTH1D) to redox processes (MT1E, MT2A) (*Figure 4E*). Gene set enrichment analysis of these results in the NE-variant cluster revealed increased expression of genes associated with activation of epithelial-mesenchymal transition ($p=6.81 \times 10^{-3}$) and pancreatic beta cell signaling ($p=1.01 \times 10^{-3}$). However, this cluster was also linked to lower levels of oxidative phosphorylation ($p=1.07 \times 10^{-25}$), possibly suggesting these subgroups of cells are biologically different (*Figure 4—figure supplement 1E*). Further examination of the ASCL1-high and NEUROD1-high clusters revealed that, across both clusters, RPM and RPM (T58A) cells were associated with higher expression of genes associated with cells in dividing phases of the cell cycle than were observed in RP cultures. We compared the proportion of G2M cells between culture-derived RPM and RP models. This revealed that RPM cells were associated with a 10- and 11-fold increase in the NE and NE-variant clusters, respectively (NE $p=9.6 \times 10^{-6}$, NE-variant $p=1.9 \times 10^{-7}$) with a relative reduction in cells in the G1 phase (*Figure 4—figure supplement 1F*). Similar trends were observed from data in human tumors (NE $p<2.2 \times 10^{-16}$, NE-variant $p=1.5 \times 10^{-13}$, *Figure 4—figure supplement 2F*).

Given the striking increase in proliferation in MYC-producing cells, we sought to understand the association between MYC and ASCL1 or NEUROD1 expression. We computed the cellular density of co-expression patterns of cells that express MYC in addition to either ASCL1 or NEUROD1. This identified that the joint density was highest for cells that co-express MYC and NEUROD1 and was localized to the NEUROD1-high cluster (*Figure 4—figure supplement 1F*). In comparison, MYC and ASCL1 co-expression patterns were heterogeneous and found in the ASCL1-high cluster as well as in cells that could not be annotated with confidence. Similar to what was previously reported by other groups, we found that ASCL1 and MYC expressions are negatively correlated (Pearson $r=-0.11$, $p=1.5 \times 10^{-4}$). Moreover, a modest correlation between MYC and NEUROD1 expression was identified in our single-cell data (Pearson $r=0.17$, $p=3.7 \times 10^{-4}$, *Figure 4—figure supplement 1G*). Given that single-cell sequencing experiments suffer from a high dropout rate, we attempted to validate MYC and NEUROD1 co-expression patterns in bulk primary and metastatic RPM tumors. Although we couldn't determine co-expression within the same cell using bulk gene expression data, we identified an extremely strong correlation between the expression of these two genes (Pearson $r=0.93$, $p=0.02$, *Figure 4—figure supplement 1H*). Taken together, we show that the overexpression of MYC (WT or T58A) may be linked to an SCLC-N phenotype.

Upon identification of the ASCL1- and NEUROD1-high clusters, we sought to further characterize how closely they resemble SCLC tumors transcriptionally using a two-pronged approach. First, we overlaid SCLC-A, SCLC-N, and SCLC-P signatures (*Chan et al., 2021*) from our single-cell data and calculated subtype enrichment scores. Enrichment patterns in RPM and RPM (T58A) xenografts revealed that both ASCL1- and NEUROD1-high clusters expressed SCLC-N-associated genes more frequently than SCLC-A and SCLC-P-associated counterparts ($p<2.2 \times 10^{-16}$, *Figure 4G*). In contrast, RP cell cultures more closely resembled cells from tumors with an SCLC-A phenotype ($p<2.2 \times 10^{-16}$, *Figure 4—figure supplement 1I*). Second, we identified the transcriptional identity of RPM xenografts and paired metastases by cellular deconvolution. Bulk gene expression of RPM tumors along

with human SCLC biopsies (*George et al., 2015*) was deconvolved into SCLC subtypes. This identified relative cellular proportions of SCLC subtypes in bulk RNA-seq data. Overall, we found high concordance with published subtype annotations and those predicted by our analysis (*Figure 4H*; *Rudin et al., 2019*). Moreover, we found that all RPM xenografts strongly exhibited an SCLC-N transcriptional program. Taken together, these data suggest that overexpression of WT or mutant MYC alters the transcriptional identity of RP cells (*Figure 4—figure supplement 2A–C*), encouraging neuroendocrine differentiation, and may assist with the transformation from a SCLC-A to SCLC-N phenotype.

## Discussion

In this report, we show that the addition of an efficiently expressed transgene encoding normal or mutant human MYC can convert weakly tumorigenic human PNEC cells, derived from a human ESC line and depleted of tumor suppressors RB1 and TP53, into highly malignant, metastatic SCLC-like cancers after implantation into the renal capsule of immunodeficient mice.

Experimental models for understanding the origins, behaviors, and control of human cancers have been developed in numerous ways for more than a century. Some of the models depend on producing tumors in animals or transforming the behavior of cells growing in culture with viruses, chemical agents, genetic changes, or other perturbations, or by deriving cells from human cancers and growing them in animals or in tissue culture. The choice of these approaches has been influenced by the goals of the research, the type of cancer under study, and available technologies.

Because SCLC is a highly aggressive cancer with a relatively stereotypic genotype that is generally recalcitrant to therapeutic strategies that have not changed appreciably in the past 30 years, we have been attempting to generate a model for SCLC in which the cancer cells are human in origin, derived from the most commonly affected pulmonary cell lineage (neuroendocrine), and equipped with the major and most common genotypic and phenotypic changes characteristic of SCLC.

The most obvious use of these cells (the RPM lines described here) is for the evaluation of the efficacy of novel therapeutic agents acting directly on tumor cells during growth in culture or during growth as xenografts in immunosuppressed mice. However, our attempts to conduct therapeutic studies with the RPM lines have not been sufficiently extensive or rigorous to warrant presentation here. In addition, treatment strategies that involve host elements other than the tumor itself, such as immune cells or other features of the tumor microenvironment, might require an experimental platform more complex than those that we have constructed.

Our choice of MYC to serve as an oncogenic driver in hESC-derived PNECs was based on both the frequent finding of high levels of *MYC* RNA in human SCLC and the use of Myc overexpression strategies to generate aggressive mouse models of SCLC (*Ireland et al., 2020*; *Mollaoglu et al., 2017*; *Huijbers et al., 2014*; *Ciampricotti et al., 2021*; *Kim et al., 2016*). However, not all human SCLC tumors have abundant *MYC* RNA, and other genes, including other members of the MYC gene family, have been implicated in the pathogenesis of SCLC. So it will be interesting to compare the ability of transgenes encoding other MYC proteins, such as NMYC and LMYC, as well as other oncoproteins (both those implicated and not implicated in human SCLC carcinogenesis) to convert RP cells into malignant tumors and to produce metastatic phenotypes, as we have observed here with RPM cells. In particular, it may be important to determine whether hESC-derived SCLC cells with varied genotypes are liable to metastasize to specific sites such as the brain and bone, which are commonly affected in patients with SCLC, but were not sites for metastatic growth in the experiments reported here. Finally, the role of other tumor suppressor genes (such as PTEN) – implicated in the development of mouse models of histological transformation of LUAD to SCLC that we have recently reported (*Gardner et al., 2024*) – should also be examined in this human ESC-derived model.

We have used transcriptional profiling as a means to compare human ESC-derived SCLC grown from RPM cells with SCLC tumor samples from patients. Other aspects of tumor cell behavior – including pathophysiological features, responses to therapeutic strategies, and the appearance of molecular, metabolic, and immunological markers – might also provide means to determine the degree of similarity between this novel stem cell-derived model and genuine human neoplasms.

Transcriptional profiling has also allowed us to conclude that the hESC-derived tumors arising from RPM RUES2 cells belong to the SCLC-N subtype, one of the major subtypes of SCLC now recognized by several laboratories and clinicians studying SCLC. It is unclear whether SCLC-N truly represents a subtype of SCLC with distinct outcomes and therapeutic sensitivities (*Gay et al., 2021*), or rather

a snapshot from a continuum in tumor evolution driven (principally) by MYC toward an aggressive, recalcitrant disease (*Choudhuri et al., 2023*; *Pongor et al., 2023*). Future studies of stem cell-derived SCLC should consider ways to generate tumors that model the other common subtypes, including SCLC-P and SCLC-I ('inflamed' SCLC) (*Gay et al., 2021*). This might be achieved by altering the cancer genotype, by using a differentiation scheme that produces physiologically varied precursor cells or by employing a wider variety of hESCs to develop SCLCs.

Finally, we note that in the experiments reported here, metastatic spread was observed from RPM tumors growing initially in the renal capsule but not from primary subcutaneous tumors initiated with the same cell culture. It may prove important to test other sites of primary tumor growth with RPM cells from the RUES2 line or with other SCLC models, to determine the degree to which metastatic spread depends on the site(s) of growth of the primary tumor and to identify specific factors that promote such spread.

# Materials and methods

## Key resources table

| Reagent type (species) or resource | Designation | Source or reference | Identifiers | Additional information |
|---|---|---|---|---|
| Cell line (human) | Embryonic stem cell – Rockefeller University Embryonic Stem Cell Line 2 (RUES2) | WiCell Research Institute, Inc | NIH approval number NIH hESC-09-0013, Registration #0013 | Passages 7–10 |
| Cell line (human) | Embryonic stem cell – ES02 (HES2) | WiCell Research Institute, Inc | NIH registry | Passages 3–7 |
| Cell line (mouse) | Mouse embryonic fibroblasts (MEF) | Global Stem | GSC-6001G | – |
| Antibody | Anti-RB1 (Clone 4H1) | Cell Signaling | Cat# 9309 | 1:500 (WB) |
| Antibody | Anti-P53 (Clone DO-1) | Santa Cruz Biotechnology | Cat# Sc-126 | 1:500 (WB) |
| Antibody | Anti-MYC (Clone D84C12) | Cell Signaling | Cat# 5605 | 1:500 (WB) |
| Antibody | Anti-HA (Clone 6E2) | Cell Signaling | Cat# 2367 | 1:500 (WB) |
| Antibody | Anti-CGRP (Clone CD8) | Sigma | Cat# c9487 | 1:150 (IHC/IF) |
| Antibody | Anti-ASCL1 (Clone 2D9) | Sigma | Cat# SAB1403577 | 1:150 (IHC/IF) |
| Antibody | Anti-NCAM1/CD56 | R&D Systems | Cat# AF2408 | 1:150 (IHC/IF) |
| Antibody | Anti-Ki67 (Clone D2H10) | Cell Signaling | Cat# 9027 | 1:150 (IHC/IF) |
| Antibody | Anti-NEUROD1 (Clone 3H8) | Sigma-Aldrich | Cat# WH004760M1 | 1:150 (IHC/IF) |
| Recombinant DNA reagent | pSLIK sh human Rb 1534 hyg (Julien Sage) | Addgene | RRID:Addgene_31500 | TET-inducible RB1 shRNA |
| Recombinant DNA reagent | FUW-tetO-hMYC (Rudolf Jaenisch) | Addgene | RRID:Addgene_20723 | TET-inducible wild-type MYC |
| Recombinant DNA reagent | pLV-tetO-myc T58A (Konrad Hochedlinger) | Addgene | RRID:Addgene_19763 | T58A mutant MYC with 3X-HA tag |
| Chemical compound, drug | Paraformaldehyde | Sigma | P6148 | Fixative |
| Chemical compound, drug | Triton X-100 | Sigma | X-100 | Permeabilization agent |
| Chemical compound, drug | Bovine serum albumin (BSA) | Life Technologies | A9647 | – |
| Chemical compound, drug | Accutase | STEMCELL Technology | 07920 | – |
| Chemical compound, drug | Y-27632 | R&D Systems | 1254 | – |
| Chemical compound, drug | Human BMP4 | R&D Systems | 314-BP-010 | – |

*Continued on next page*

*Continued*

| Reagent type (species) or resource | Designation | Source or reference | Identifiers | Additional information |
|---|---|---|---|---|
| Chemical compound, drug | Human bFGF | R&D Systems | 233-FB | – |
| Chemical compound, drug | Human Activin A | R&D Systems | 338-AC | – |
| Chemical compound, drug | Dorsomorphin dihydrochloride | Sigma | AMBH2D6FB2F5 | – |
| Chemical compound, drug | SB431542 | R&D Systems | TB1614-GMP | – |
| Chemical compound, drug | IWP2 | Sigma | I0536 | – |
| Chemical compound, drug | CHIR99021 | R&D Systems | 5439-DK | – |
| Chemical compound, drug | Human FGF10 | R&D Systems | Qk003-0050 | – |
| Chemical compound, drug | Human FGF7 | R&D Systems | 251-KG | – |
| Chemical compound, drug | All-trans retinoic acid (ATRA) | Sigma | 554720 | – |
| Chemical compound, drug | Dexamethasone | Sigma | 265005 | – |
| Chemical compound, drug | 8-Bromo-cAMP | Sigma-Aldrich | TA9H9A9A7499 | – |
| Chemical compound, drug | 3,7-Dihydro-1-methyl-3-(2-methylpropyl)-1*H*-purine-2,6-dione (IBMX) | Sigma-Aldrich | TA9H98DB18B1 | – |
| Chemical compound, drug | DAPT | Sigma-Aldrich | D5942 | – |
| Chemical compound, drug | Monothioglycerol | Sigma | M6145 | |
| Other | mTeSR1 | STEMCELL Technologies | 85850 | hESC maintenance for transduced hESC culture |
| Other | Matrigel | Corning | 354277 | ECM-coated plates for maturation |
| Other | DMEM/F-12 | Gibco | 10565-018 | Medium |
| Other | N2 | Gibco | 17502001 | Supplement |
| Other | B27 | Gibco | 17504044 | Supplement |
| Other | Ascorbic acid | Sigma | 1043003 | |
| Other | GlutaMAX | Life Technologies | 35050061 | Supplement |
| Other | 0.05% Trypsin/0.02% EDTA | Life Technologies | 25300062 | Enzyme |

## Generation of lung cells, including PNECs, from hESCs

Protocols for maintenance of hESCs and generation of PNECs and other lung cells were modified from previous studies (*Huang et al., 2015*; *Huang et al., 2014*; *Chen et al., 2019*). Two hESC lines, RUES2 (Rockefeller University Embryonic Stem Cell Line 2, NIH approval number NIH hESC-09-0013, Registration number 0013; passages 7–10) and ES02 (HES2, NIH registry, WiCell Research Institute, Inc passages 3–7) were cultured and maintained in an undifferentiated state on irradiated mouse embryonic fibroblasts (Global Stem, cat. no. GSC-6001G). All cells were purchased from ATCC or WiCell in the past 3 years, and tests for mycoplasma were negative. The hESC lines were regularly checked for chromosome abnormalities and maintained with normal chromosome numbers. All embryonic stem

cell studies were approved by the Institutional Review Board (IRB) at the University of Chicago, or by the Tri-Institutional ESCRO committee under protocols 2016-01-002 and 2017-10-004 (Weill Cornell Medicine, Memorial Sloan Kettering Cancer Center, and Rockefeller University).

hESC differentiation into endoderm was performed in serum-free differentiation (SFD) media of DMEM/F12 (3:1) (Life Technologies) supplemented with N2 (Life Technologies), B27 (Life Technologies), ascorbic acid (50 µg/ml, Sigma), GlutaMAX (2 mM, Life Technologies), monothioglycerol (0.4 µM, Sigma), 0.05% bovine serum albumin (BSA) (Life Technologies) at 37°C in a 5% $CO_2$/5% $O_2$/90% $N_2$ environment. hESCs were treated with Accutase (STEMCELL Technologies) and plated onto low attachment six-well plates (Corning Incorporated, Tewksbury MA, USA), resuspended in endoderm induction media containing Y-27632 (10 µM), human BMP4 (0.5 ng/ml), human bFGF (2.5 ng/ml), and human Activin A (100 ng/ml) for 72–84 hr dependent on the formation rates of endoderm cells. On day 3 or 3.5, the embryoid bodies were dissociated into single cells using 0.05% Trypsin/0.02% EDTA and plated onto fibronectin-coated (Sigma), 24-well tissue culture plates (~100,000–150,000 cells/well). For induction of anterior foregut endoderm, the endoderm cells were cultured in SFD medium supplemented with 1.5 µM dorsomorphin dihydrochloride and 10 µM SB431542 for 36–48 hr, and then switched to 36–48 hr of 10 µM SB431542 and 1 µM IWP2 treatment.

For induction of early-stage lung progenitor cells (days 6–15), the resulting anterior foregut endoderm was treated with CHIR99021 (3 µM), human FGF10 (10 ng/ml), human FGF7 (10 ng/ml), human BMP4 (10 ng/ml), and all-trans retinoic acid (ATRA; 50 nM) in SFD medium for 8–10 days. Day 10–15 cultures were maintained in a 5% $CO_2$/air environment. On days 15 and 16, the lung field progenitor cells were re-plated after brief 1 min trypsinization onto fibronectin-coated plates, in the presence of SFD containing either a combination of five factors: CHIR99021 (3 µM), human FGF10 (10 ng/ml), human FGF7 (10 ng/ml), human BMP4 (10 ng/ml), and ATRA (50 nM), or three factors: CHIR99021 (3 µM), human FGF10 (10 ng/ml), and human FGF7 (10 ng/ml) for days 14–16. Day 16–25 cultures of late-stage lung progenitor cells were maintained in SFD media containing CHIR99021 (3 µM), human FGF10 (10 ng/ml), and human FGF7 (10 ng/ml) in a 5% $CO_2$/air environment.

For differentiation of mature lung cells (days 25–55), cultures were re-plated after brief trypsinization onto 3.3% Matrigel-coated 24-well plates in SFD media containing maturation components CHIR99021 (3 µM), human FGF10 (10 ng/ml), human FGF7 (10 ng/ml), and DCI (50 nM dexamethasone, 0.1 mM 8-bromo-cAMP and 0.1 mM IBMX [3,7-dihydro-1-methyl-3-(2-methylpropyl)-1$H$-purine-2,6-dione]). DAPT (10 µM) was added to the maturation media for induction of PNECs. All growth factors and most small molecules used for hESC differentiation were purchased from R&D Systems. ATRA, 8-bromo-cAMP, IBMX, and DAPT were purchased from Sigma-Aldrich.

## Lentivirus transduction of hESCs

HSC lines were constructed to contain DOX-inducible cassettes encoding shRNAs specific for the TP53 and RB1 tumor suppressor genes (RP cells) and RP cells containing DOX-inducible cassettes encoding WT human MYC or a stable, mutant version of MYC (T58A), to generate RPM and RPM (T58A) cells. The lentiviral vector expressing tetracycline (TET)-inducible shRNA against human P53 was purchased from Gentarget. Inc (cat# LVP-343-RB-PBS). The lentiviral vectors expressing TET-inducible shRNAs against human RB1 construct ('pSLIK sh human Rb 1534 hyg' was a gift from Julien Sage; plasmid # 31500) (*Conklin et al., 2012*), TET-inducible WT MYC (FUW-tetO-hMYC was a gift from Rudolf Jaenisch; plasmid # 20723) or mutant MYC (T58A) tagged at the N-terminus with three copies of a hemagglutinin tag (3X-HA) (pLV-tetO-myc T58A was a gift from Konrad Hochedlinger; plasmid # 19763) were obtained from Addgene and sequence-verified prior to use. The shRNA target sequences are as follows: RB1 # 1: 5′-GGACATGTGAACTTATATA-3′, RB1 # 2: 5′-GAACGATTATCCATTCAAA-3′, p53: 5′-CACCATCCACTACAACTACAT-3′.

To generate lentiviral particles, the above plasmids were transfected into HEK293T cells with the PC-Pack2 lentiviral packaging mix (Cellecta, Inc), according to the manufacturer's protocol. High titer viral particles were used to transduce hESCs in serum-free conditions, and antibiotic selection of transduced hESCs was performed without MEF feeder cells, using mTeSR1 stem cell media (STEMCELL Technologies). The efficiency of RB1 or P53 knockdown and production of MYC or MYC (T58A) was determined by western blot after antibiotic selection using the following antibodies: anti-RB1 (Cell Signaling, clone 4H1, Cat# 9309), anti-P53 (Santa Cruz, clone DO-1, Cat# Sc-126), anti-MYC (Cell Signaling, clone D84C12, Cat# 5605), anti-HA (Cell Signaling, clone 6E2, Cat# 2367), and anti-GAPDH

(Cell Signaling, clone 14C10, Cat# 2118). All antibodies for western blot were used at a 1:500 working dilution.

## Immunohistochemistry

Living cells in culture were directly fixed in 4% paraformaldehyde for 25 min, followed with 15 min permeabilization in 1% Triton X-100. Histology on tissues from mice was performed on paraffin-embedded or frozen sections from xenografted tumors and corresponding normal tissues, as previously described (*Chen et al., 2015*). Tissues were either fixed overnight in 10% buffered formalin and transferred to 70% ethanol, followed by paraffin embedding, or snap-frozen in OCT medium (Fisher Scientific, Pittsburgh, PA, USA) and fixed in 10% buffered formalin, followed by paraffin embedding. For immunofluorescence, cells or tissue sections were immunostained with antibodies and counterstained with 4,6-diamidino-2-phenylindole. Adjacent sections stained with H&E were used for comparison.

Antibodies used for immunostaining or western blot experiments are as follows: anti-CGRP (Sigma, Clone CD8, Cat# c9487), anti-ASCL1 (Sigma, clone 2D9, Cat# SAB1403577), anti-NCAM1/CD56 (R&D Systems, cat# AF2408), anti-Ki67 (Cell Signaling, clone D2H10, Cat# 9027), anti-MYC (Abcam, clone Y69, Cat# ab32072), and anti-NEUROD1 (Sigma-Aldrich, clone 3H8, Cat# WH004760M1). All antibodies were used at a 1:150 working dilution.

## Fluorescent-activated cell sorting

Fluorescent-activated cell sorting (FACS) to detect definitive endoderm cells was performed using APC anti-c-KIT (Invitrogen, Cat# CD11705), PE anti-human CXCR4 (BioLegend, Clone 12G5, Cat# 306506), and APC anti-EpCAM (Invitrogen, Clone G8.8, Cat# 17-5791-80) at 1:100 dilution. Cells were incubated with primary antibodies for 30 min at 4°C, then washed and suspended in 0.1% BSA/PBS buffer. PE and APC filters were then used to detect cells double positive for KIT and CXCR4, or EpCAM and CXCR4 by signal intensity gating. FACS with anti-CGRP antibody (Abcam, clone 4902, Cat# ab81887) was used to detect CGRP+ cells. Cells were first incubated with anti-human CGRP antibody for 30 min at room temperature, followed by incubation (1:1000) of secondary antibody conjugated with R-phycoerythrin (PE) for 30 min at room temperature. Then, cells were washed and suspended in 0.1% BSA/PBS buffer. PE filter was then used to separate cells into CGRP+ and CGRP- subgroups by signal intensity gating. Negative controls stained with isotype controls were performed with sample measurements. Cells were sorted on a BD FACSAria II, and data were analyzed in FlowJo.

## Xenograft formation

0.3–0.5 × 10$^6$ hESC-derived lung cells or PNECs (at day 55 with or without 30 days of prior exposure to 10 μM DAPT alone or to 10 μM DAPT and 1 μM DOX to reduce P53 and RB1, and also induce MYC expression) were injected subcutaneously or under the renal capsule of 6- to 8-week-old, female NOD. Cg-Prkdc$^{scid}$ Il2rgtm1WjI$^{tm1WjI}$/SzJ (NSG) mice (Jackson Laboratory, Bar Harbor, ME) (*Chen et al., 2012*). DOX diet began the day after injection (Teklad; 625 ppm); tumor development was monitored at least two to three times weekly. When animals became moribund or tumor size reached protocol limitations, mice were sacrificed, necropsy performed, and tumors harvested for further histological or molecular study.

## Single-cell sequencing and data analysis

Single-cell capture, reverse transcription, cell lysis, and library preparation were carried out by the WCM Epigenomics Core Facility using Chromium Single Cell 3' v3 kits according to the manufacturer's protocol (10x Genomics, USA). Single-cell suspensions of RUES2-derived RPM and RPM (T58A) day 50 cultures were prepared by incubating wells in 0.05% Trypsin/0.02% EDTA for 10–15 min. Digestions were then quenched with excess 0.1% BSA/PBS buffer and passed through a 40 μm mesh strainer. EGFP+ singlets were sorted at the WCM Flow Cytometry Core Facility on a BD Influx or Sony MA900 cell sorter. Each RUES2-derived RPM and RPM (T58A) day 50 culture pools were isolated from three wells of a 12-well plate across two biological replicates, targeting ~50,000 viable, EGFP+ viable cells per replicate. Cell counts were then adjusted to 6000–9000 cells/50 μl in PBS by Trypan Blue exclusion to achieve an estimated capture of 4000–5000 cells. Sequencing was performed by the WCM Genomics Resources Core Facility on a HiSeq 2500 (Illumina, paired-end protocol with 26 base pairs

for read 1 and 98 bases for read 2). Single-cell RNA-seq from RP cultures was obtained from previous experiments (*Chen et al., 2019*). Transcript abundance was quantified by mapping sequencing reads to a custom human transcriptome reference (GENCODE v42, hg38) that included EGFP and was performed using simpleaf/alevin-fry (*He and Patro, 2023*).

Single-cell analyses, including quality filtering, integration, clustering, differential gene expression, and reduced-dimensionality visualization, were performed using the Seurat package in R as described in the package SCTransform tutorial (version 4.3.0) (*Hao et al., 2021*). Briefly, cells to be included in the analysis were required to have at least 4000 and at most 45,000 unique molecular identifiers and expressed between 1000 and 7500 unique genes. In addition, cells were excluded if more than 10% of the determined RNA sequences mapped to mitochondrial genes or if they were classified as doublets by scds (*Bais and Kostka, 2020*). In total, 7728 cells from two RP culture samples, 1875 cells from RPM cultures, 1562 cells from RPM (T58A) cultures, 3401 cells from RPM tumors, and 3463 cells from RPM (T58A) tumors passed these filters for quality. Samples were integrated using Harmony with the top 15 principal components and were run with up to 50 iterations of the algorithm (*Korsunsky et al., 2019*). Dimensionality reduction was achieved by creating a Uniform Manifold Approximation and Projection of the first 15 dimensions of harmony integration. Clustering resolution was set at 0.1 for the integrated dataset, resulting in three clusters. Subclustering was conducted in a similar manner and involved re-integration of cells from the neuroendocrine cluster and used identical parameters. Cell-type annotation was called using the Azimuth functionality within the Seurat package, followed by manual inspection, and enrichment analysis was conducted using enrichR (*Chen et al., 2013*). A lung atlas containing 584,884 cells was obtained for Azimuth-mediated reference mapping from the HubMap portal (*Bais and Kostka, 2020*; *Deprez et al., 2020*; *Goldfarbmuren et al., 2020*; *Habermann et al., 2020*; *Madissoon et al., 2019*; *Morse et al., 2019*; *Schupp et al., 2021*; *Travaglini et al., 2020*; *Vieira Braga et al., 2019*).

## Bulk RNA-seq of the xenograft tumors

Total RNA from the primary and metastatic tumors was isolated using Direct-zol RNA Miniprep kit (Zymo Research). Following RNA isolation, total RNA integrity was checked using a 2100 Bioanalyzer (Agilent Technologies). RNA concentrations were measured using the NanoDrop system (Thermo Fisher Scientific). Preparation of RNA sample library and RNA-seq were performed by the Genomics Core Laboratory at Weill Cornell Medicine or Northwestern University. Messenger RNA was prepared using the TruSeq Stranded mRNA Sample Library Preparation kit (Illumina), according to the manufacturer's instructions. The normalized libraries were pooled and sequenced on Illumina NovaSeq 6000 sequencer with paired-end 50 cycles. Sequencing adapters were trimmed, and quality metrics were generated using fastp (*Chen et al., 2018*). Transcript abundance was quantified by mapping sequencing reads to a reference transcriptome (GENCODE v42, hg38) using Salmon (*Patro et al., 2017*).

## Differential gene expression and enrichment analysis

Identification of differentially expressed genes involved adjusting for sample identity and MYC expression status when appropriate and was achieved using model-based analysis of single-cell transcriptomics (*Finak et al., 2015*). Gene set enrichment analysis was conducted by mapping differential expression results to hallmark pathways using clusterProfiler (version 4.0) (*Liberzon et al., 2015*; *Wu et al., 2021*). Cell-level enrichment scores for top 20 cluster markers were previously identified (*Chan et al., 2021*) using the AddModuleScore functionality of Seurat. Patterns of MYC and NEUROD1 co-expression were limited to cells in which at least one MYC and one NEUROD1 transcript were detected post-processing.

## Deconvolution of bulk RNA-seq xenografts into SCLC subtypes

Processed single-cell RNA-seq data of primary SCLC were obtained from the cellxgene portal (*Chan et al., 2021*). Bulk RNA-seq data for human SCLC tumors (reads per kilobase per million mapped reads) were obtained from cBioPortal. Subtype annotations for 45 patients were obtained from previously published classifications (*Rudin et al., 2019*). Deconvolution of bulk RNA-seq data was achieved through BayesPrism (version 2.0) (*Chu et al., 2022*). Briefly, single-cell transcriptomes from SCLC-A, SCLC-N, SCLC-P, and lung adenocarcinoma (LUAD; 'NSCLC') samples were annotated as tumor cells.

For deconvolution, aggregated tumor annotations were provided as cell-type labels, and specific tumor types (e.g. SCLC-A) were provided as tumor states. BayesPrism received inputs from single-cell annotations and associated count matrix, xenograft, and primary tumor bulk RNA-seq data and was run with outlier.cut=0.01 and outlier.fraction=0.1. Analysis of resultant cell-type fractions was limited to SCLC-A, SCLC-N, and SCLC-P. Estimated cell-type fractions were converted to proportion metrics within the three SCLC subtypes and visualized for concordance with published annotations.

## Statistical analyses

Group sample sizes for all in vivo experiments were estimated based off of our previous studies, with at least five animals per experimental arm being included for purposes of statistical testing (*Huang et al., 2015*; *Huang et al., 2014*; *Chen et al., 2019*; *Chen et al., 2016*; *Unni et al., 2015*). For mouse experiments, animals were randomly assigned to each group, and no animals were excluded from the analyses. All statistical tests are two-sided. No adjustments were made for multiple comparisons. The relevant investigators (HJC, EEG, and KZ) were blinded to experimental allocations among different experimental arms. For all parametric statistical analyses, data were determined to be normally distributed by the D'Agostino-Pearson test. For comparison between experimental and control groups at a specific time point or tissue site in *Figures 2 and 4*, two-sided Student's t-tests, one-way or two-way ANOVA tests, Fisher's exact tests, and two-sided Kolmogorov-Smirnov tests were used in GraphPad Prism. Calculated p-values at or below $2.2 \times 10^{-16}$ were reported as $p < 2.2 \times 10^{-16}$, in line with statistical reporting in R.

## Acknowledgements

We thank Oksana Mashadova and Sukanya Goswami in the Varmus Laboratory for technical support, Arun Unni and John Ferrarone in the Varmus Laboratory, and Jonathan Chen in the Chen Laboratory for useful advice. This study was supported by awards to HV and OE from the US Department of Defense (LC160136) and the US National Cancer Institute (U01CA224326), funds from the Meyer Cancer Center, Weill Cornell Medicine (to HV), and a K99/R00 NIH Pathway to Independence Award (4R00CA226353), US Department of Defense (RA220012), and a Lung Cancer Research Foundation (LCRF) Pilot Project Award (to HJC).

## Additional information

### Funding

| Funder | Grant reference number | Author |
|---|---|---|
| Department of Defense Education Activity | LC160136 | Olivier Elemento Harold Varmus |
| National Cancer Institute | U01CA224326 | Harold Varmus |
| National Institutes of Health | 4R00CA226353 | Huanhuan Joyce Chen |
| Department of Defense Education Activity | RA220012 | Huanhuan Joyce Chen |
| Lung Cancer Research Foundation | Pilot Project | Huanhuan Joyce Chen |

The funders had no role in study design, data collection and interpretation, or the decision to submit the work for publication.

### Author contributions

Huanhuan Joyce Chen, Conceptualization, Data curation, Formal analysis, Funding acquisition, Investigation, Methodology, Writing – original draft, Writing – review and editing; Eric E Gardner, Data curation, Formal analysis, Validation, Investigation, Methodology, Writing – original draft, Writing – review and editing; Yajas Shah, Resources, Formal analysis, Visualization, Writing – review and editing; Kui Zhang, Formal analysis, Validation, Writing – review and editing; Abhimanyu Thakur, Formal analysis,

Visualization, Writing – review and editing; Chen Zhang, Resources, Formal analysis, Methodology; Olivier Elemento, Resources, Supervision, Funding acquisition, Writing – review and editing; Harold Varmus, Conceptualization, Supervision, Funding acquisition, Writing – original draft, Project administration, Writing – review and editing

**Author ORCIDs**
Huanhuan Joyce Chen  https://orcid.org/0000-0002-1446-7184
Eric E Gardner  https://orcid.org/0000-0002-1552-2675
Abhimanyu Thakur  http://orcid.org/0000-0002-0846-0998

**Ethics**
At the time of manuscript submission, all animals were handled, and experiments conducted according to approved institutional animal care and use committee (IACUC) protocols (#2015-0017) of Weill Cornell Medicine (WCM).

Reviewer #1 (Public review): https://doi.org/10.7554/eLife.93170.3.sa1
Reviewer #3 (Public review): https://doi.org/10.7554/eLife.93170.3.sa2
Author response https://doi.org/10.7554/eLife.93170.3.sa3

---

# Additional files

## Supplementary files
MDAR checklist

Supplementary file 1. Supplementary tables. (**a**) Differentially expressed transcripts used to identify major cell type clusters in single cell RNA-seq data. (**b**) Differentially expressed transcripts from neuroendocrine component sub-clustering and groupings. (**c**) Differentially expressed transcripts used to compare NE and NE-variant sub-clustering.

Source code 1. Source code file.

## Data availability
Sequencing data generated in this study are available at the NCBI Gene Expression Omnibus (GEO) under accession GSE255757 (samples SRR27965726-SRR27965731) and the Sequence Read Archive (SRA) under BioProject SUB15343548 (samples SAMN48725919-SAMN48725923). The dataset has been deposited in the NCBI Sequence Read Archive (SRA) under BioProject SUB15343548, now registered as PRJNA1267586 (2025).

The following datasets were generated:

| Author(s) | Year | Dataset title | Dataset URL | Database and Identifier |
|---|---|---|---|---|
| Shah Y, Gardner E, Chen HJ, Varmus H | 2025 | Formation of malignant, metastatic small cell lung cancers through overproduction of cMYC protein in TP53 and RB1 depleted pulmonary neuroendocrine cells derived from human embryonic stem cells | https://www.ncbi.nlm.nih.gov/geo/query/acc.cgi?acc=GSE255757 | NCBI Gene Expression Omnibus, GSE255757 |
| Shah Y, Gardner E, Chen HJ, Varmus H | 2025 | Formation of malignant, metastatic small cell lung cancers through overproduction of cMYC protein in TP53 and RB1 depleted pulmonary neuroendocrine cells derived from human embryonic stem cells | https://www.ncbi.nlm.nih.gov/bioproject/PRJNA1267586 | NCBI BioProject, PRJNA1267586 |

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
