## [Editor Report · eLife Assessment]

Given a great need for novel human model systems to study small cell lung cancer (SCLC), the authors describe an **important** pre-clinical model with broad potential for the study of how genetic perturbations or drug treatments alter SCLC tumor growth, metastasis, and response to therapy. For the major finding, the authors provide **convincing** evidence that RB/TP53 suppression coupled with MYC overexpression in an ES cell-derived model system results in aggressive and metastatic SCLC. However, the impact of the work would have been increased with the inclusion of a broader set of genetic perturbations, such as over-expression of MYCL, to better model major SCLC phenotypes. The new model described will be of significant interest to researchers studying lung cancer.

---

## [Referee Report · Reviewer #1 (Public review)]

Summary:

The authors introduced their previous paper with the concise statement that "the relationships between lineage-specific attributes and genotypic differences of tumors are not understood" (Chen et al., JEM 2019, PMID: 30737256). For example, it is not clear why combined loss of RB1 and TP53 is required for tumorigenesis in SCLC or other aggressive neuroendocrine (NE) cancers, or why the oncogenic mutations in KRAS or EGFR that drive NSCLC tumorigenesis are found so infrequently in SCLC. This is the main question addressed by the previous and current papers.

One approach to this question is to identify a discrete set of genetic/biochemical manipulations that are sufficient to transform non-malignant human cells into SCLC-like tumors. One group reported transformation of primary human bronchial epithelial cells into NE tumors through a complex lentiviral cocktail involving inactivation of pRB and p53 and activation of AKT, cMYC and BCL2 (PARCB) (Park et al., Science 2018, PMID: 30287662). The cocktail previously reported by Chen and colleagues to transform human pluripotent stem-cell (hPSC)-derived lung progenitors (LPs) into NE xenografts was more concise: DAPT to inactivate NOTCH signaling combined with shRNAs against RB1 and TP53. However, the resulting RP xenografts lacked important characteristics of SCLC. Unlike SCLC, these tumors proliferated slowly and did not metastasize, and although small subpopulations expressed MYC or MYCL, none expressed NEUROD1.

MYC is frequently amplified or expressed at high levels in SCLC, and here, the authors have tested whether inducible expression of MYC could increase the resemblance of their hPSC-derived NE tumors to SCLC. These RPM cells (or RPM T58A with stabilized cMYC) engrafted more consistently and grew more rapidly than RP cells, and unlike RP cells, formed liver metastases when injected into the renal capsule. Gene expression analyses reveled that RPM tumor subpopulations expressed NEUROD1, ASCL1 and/or YAP1.

The hPSC-derived RPM model is a major advance over the previous RP model. This may become a powerful tool for understanding SCLC tumorigenesis and progression and for discovering gene dependencies and molecular targets for novel therapies. However, the specific role of cMYC in this model needs to be clarified.

Recommended Revision:

cMYC can drive proliferation, tumorigenesis or apoptosis in a variety of lineages depending on concurrent mutations. For example, in the Park et al., study, normal human prostate cells could be reprogrammed to form adenocarcinoma-like tumors by activation of cMYC and AKT alone, without manipulation of TP53 or RB1. In their previous manuscript, the authors carefully showed the role of each molecular manipulation in NE tumorigenesis. DAPT was required for NE differentiation of LPs to PNECs, shRB1 was required for expansion of the PNECs, and shTP53 was required for xenograft formation. cMYC expression could influence each of these steps, and importantly, could render some steps dispensable. For example, shRB1 was previously necessary to expand the DAPT-induced PNECs, as neither shTP53 nor activation of KRAS or EGFR had no effect on this population, but perhaps cMYC overexpression could expand PNECs even in the presence of pRB, or even induce LPs to become PNECs without DAPT. Similarly, both shRB1 and shTP53 were necessary for xenograft formation, but maybe not if cMYC is overexpressed. If a molecular hallmark of SCLC, such as loss of RB1 or TP53, has become dispensable with the addition of cMYC, this information is critically important in interpreting this as a model of SCLC tumorigenesis.

To interpret the role of cMYC expression in hPSC-derived RPM tumors, we need to know what this manipulation does without manipulation of pRB, p53 or NOTCH, alone or in combination. There are 7 relevant combinations that should be presented in this manuscript: (1) cMYC alone in LPs, (2) cMYC + DAPT, (3) cMYC + shRB1, (4) cMYC + DAPT + shRB1, (5) cMYC + shTP53, (6) cMYC + DAPT + shTP53, and (7) cMYC + shRB1 + shTP53. Wild-type cMYC is sufficient; further exploration with the T58A mutant would not be necessary.

Please present the effects of these combinations on LP differentiation to PNECs, expansion of PNECs as well as other lung cells, xenograft formation and histology, and xenograft growth rate and capacity for metastasis. If this could be clarified experimentally, and the results discussed in the context of other similar approaches such as the Park et al., paper, this study would be a major addition to the field.

---

## [Referee Report · Reviewer #3 (Public review)]

This revision and the accompanying rebuttal indicates the authors want to publish their studies without providing several of the reviewer requested additional experiments (such as determining the impact of other Myc family members on metastatic behavior and expression characteristics compared to overexpression of c-Myc), and determining whether the tumors were responsive or not to standard clinically used therapies. Their argument is the author team has moved on to other endeavors, it is important to communicate their findings to the research field, and they have indicated these issues in the Discussion. All of these things are reasonable. However, there two things that would help. The first is to have the authors clearly state in the Discussion section "Limitations of the current study" and then list these out. In the current format the indication that the authors recognize the "limitations" is not clearly stated. An example - of such a limitation is how well their model now provides a human SCLC like tumor that metastasizes. We know that in patients SCLC is widely metastatic, but in SCLC patient derived xenografts with subcutaneous injection that is not seen, so if their model now generated widely metastatic behavior like that seen in patients, this report and the associated resources would be a significant advance to the field. However, their data shows that using their model the subcutaneous tumors don't metastasize, and even with renal capsule models metastases are not common and do not go to important sites (e.g. brain). Second, a major reason for publishing this paper is that their model system would be available as a resource for the field to study. However, I could not find in the paper or the Methods section any statement as to the availability of this presumable important resource. If the resources will not be easily available in a format that others can readily study (e.g. with instructions on how to handle the cells which would seem to be more complicated than other patient derived SCLC models) then of course the value of this paper to the field as a whole is dramatically reduced. I would assume the authors want their model to be used by other investigators and thus a clear statement of model availability and how to routinely handle their model is important to include in their manuscript.

---

## [Author Response]

The following is the authors’ response to the previous reviews.

**Public Reviews:**

**Reviewer #1 (Public Review):**
Summary:The authors introduced their previous paper with the concise statement that "the relationships between lineage-specific attributes and genotypic differences of tumors are not understood" (Chen et al., JEM 2019, PMID: 30737256). For example, it is not clear why combined loss of RB1 and TP53 is required for tumorigenesis in SCLC or other aggressive neuroendocrine (NE) cancers, or why the oncogenic mutations in KRAS or EGFR that drive NSCLC tumorigenesis are found so infrequently in SCLC. This is the main question addressed by the previous and current papers.One approach to this question is to identify a discrete set of genetic/biochemical manipulations that are sufficient to transform non-malignant human cells into SCLC-like tumors. One group reported the transformation of primary human bronchial epithelial cells into NE tumors through a complex lentiviral cocktail involving the inactivation of pRB and p53 and activation of AKT, cMYC, and BCL2 (PARCB) (Park et al., Science 2018, PMID: 30287662). The cocktail previously reported by Chen and colleagues to transform human pluripotent stem-cell (hPSC)-derived lung progenitors (LPs) into NE xenografts was more concise: DAPT to inactivate NOTCH signaling combined with shRNAs against RB1 and TP53. However, the resulting RP xenografts lacked important characteristics of SCLC. Unlike SCLC, these tumors proliferated slowly and did not metastasize, and although small subpopulations expressed MYC or MYCL, none expressed NEUROD1.MYC is frequently amplified or expressed at high levels in SCLC, and here, the authors have tested whether inducible expression of MYC could increase the resemblance of their hPSC-derived NE tumors to SCLC. These RPM cells (or RPM T58A with stabilized cMYC) engrafted more consistently and grew more rapidly than RP cells, and unlike RP cells, formed liver metastases when injected into the renal capsule. Gene expression analyses revealed that RPM tumor subpopulations expressed NEUROD1, ASCL1, and/or YAP1.The hPSC-derived RPM model is a major advance over the previous RP model. This may become a powerful tool for understanding SCLC tumorigenesis and progression and for discovering gene dependencies and molecular targets for novel therapies. However, the specific role of cMYC in this model needs to be clarified.cMYC can drive proliferation, tumorigenesis, or apoptosis in a variety of lineages depending on concurrent mutations. For example, in the Park et al., study, normal human prostate cells could be reprogrammed to form adenocarcinoma-like tumors by activation of cMYC and AKT alone, without manipulation of TP53 or RB1. In their previous manuscript, the authors carefully showed the role of each molecular manipulation in NE tumorigenesis. DAPT was required for NE differentiation of LPs to PNECs, shRB1 was required for expansion of the PNECs, and shTP53 was required for xenograft formation. cMYC expression could influence each of these steps, and importantly, could render some steps dispensable. For example, shRB1 was previously necessary to expand the DAPT-induced PNECs, as neither shTP53 nor activation of KRAS or EGFR had no effect on this population, but perhaps cMYC overexpression could expand PNECs even in the presence of pRB, or even induce LPs to become PNECs without DAPT. Similarly, both shRB1 and shTP53 were necessary for xenograft formation, but maybe not if cMYC is overexpressed. If a molecular hallmark of SCLC, such as loss of RB1 or TP53, has become dispensable with the addition of cMYC, this information is critically important in interpreting this as a model of SCLC tumorigenesis.

The reviewer’s suggestion may be possible; indeed, in a recent report from our group (Gardner EE, et al., Science 2024) we have shown, using genetically engineered mouse modeling coupled with lineage tracing, that the cMyc oncogene can selectively expand Ascl1+ PNECs in the lung.

We agree with the reviewer that not having a better understanding of the individual components necessary and/or sufficient to transform hESC-derived LPs is an important shortcoming of this current work. However, we would like to stress three important points about the comments: (1) tumors were reviewed and the histological diagnoses were certified by a practicing pulmonary pathologist at WCM (our co-author, C. Zhang); (2)the observed transcriptional programs were consistent with primary human SCLC; and (3) RB1-proficient SCLC is now recognized as a rare presentation of SCLC (Febrese-Aldana CA, et al., Clin. Can. Res. 2022. PMID: 35792876).

To interpret the role of cMYC expression in hPSC-derived RPM tumors, we need to know what this manipulation does without manipulation of pRB, p53, or NOTCH, alone or in combination. Seven relevant combinations should be presented in this manuscript: (1) cMYC alone in LPs, (2) cMYC + DAPT, (3) cMYC + shRB1, (4) cMYC + DAPT + shRB1, (5) cMYC + shTP53, (6) cMYC + DAPT + shTP53, and (7) cMYC + shRB1 + shTP53. Wildtype cMYC is sufficient; further exploration with the T58A mutant would not be necessary.

We respectfully disagree that an interrogation of the differences between the phenotypes produced by wildtype and Myc(T58A) would not be informative. (Our view is confirmed by the second reviewer; see below.) It is well established that Myc gene or protein dosage can have profound effects on in vivo phenotypes (Murphy DJ, et al., Cancer Cell 2008. PMID: 19061836). The “RPM” model of variant SCLC developed by Trudy Oliver’s lab relied on the conditional T58A point mutant of cMyc, originally made by Rob Wechsler-Reya. While we do not discuss the differences between Myc and Myc(T58A), it is nonetheless important to present our results with both the WT and mutant MYC constructs, as we are aware of others actively investigating differences between them in GEMM models of SCLC tumor development.

We agree with the reviewer about the virtues of trying to identify the effects of individual gene manipulations; indeed our original paper (Chen et al., J. Expt. Med. 2019), describing the RUES2derived model of SCLC did just that, carefully dissecting events required to transform LPs towards a SCLC-like state. The central purpose of the current study was to determine the effects of adding cMyc on the behavior of weakly tumorigenic SCLC-like cells cMyc. Presenting data with these two alleles to seek effects of different doses of MYC protein seems reasonable.

This reviewer considers that there should be a presentation of the effects of these combinations on LP differentiation to PNECs, expansion of PNECs as well as other lung cells, xenograft formation and histology, and xenograft growth rate and capacity for metastasis. If this could be clarified experimentally, and the results discussed in the context of other similar approaches such as the Park et al., paper, this study would be a major addition to the field.
**Reviewer #2 (Public Review):**
Summary:Chen et al use human embryonic stem cells (ESCs) to determine the impact of wildtype MYC and a point mutant stable form of MYC (MYC-T58A) in the transformation of induced pulmonary neuroendocrine cells (PNEC) in the context of RB1/P53 (RP) loss (tumor suppressors that are nearly universally lost in small cell lung cancer (SCLC)). Upon transplant into immune-deficient mice, they find that RP-MYC and RP-MYC-T58A cells grow more rapidly, and are more likely to be metastatic when transplanted into the kidney capsule, than RP controls. Through single-cell RNA sequencing and immunostaining approaches, they find that these RPM tumors and their metastases express NEUROD1, which is a transcription factor whose expression marks a distinct molecular state of SCLC. While MYC is already known to promote aggressive NEUROD1+ SCLC in other models, these data demonstrate its capacity in a human setting that provides a rationale for further use of the ESC-based model going forward. Overall, these findings provide a minor advance over the previous characterization of this ESC-based model of SCLC published in Chen et al, J Exp Med, 2019.

We consider the findings more than a “minor” advance in the development of the model, since any useful model for SCLC would need to form aggressive and metastatic tumors.

The major conclusion of the paper is generally well supported, but some minor conclusions are inadequate and require important controls and more careful analysis.Strengths:(1) Both MYC and MYC-T58A yield similar results when RP-MYC and RP-MYCT58A PNEC ESCs are injected subcutaneously, or into the renal capsule, of immune-deficient mice, leading to the conclusion that MYC promotes faster growth and more metastases than RP controls.(2) Consistent with numerous prior studies in mice with a neuroendocrine (NE) cell of origin (Mollaoglu et al, Cancer Cell, 2017; Ireland et al, Cancer Cell, 2020; Olsen et al, Genes Dev, 2021), MYC appears sufficient in the context of RB/P53 loss to induce the NEUROD1 state. Prior studies also show that MYC can convert human ASCL1+ neuroendocrine SCLC cell lines to a NEUROD1 state (Patel et al, Sci Advances, 2021); this study for the first time demonstrates that RB/P53/MYC from a human neuroendocrine cell of origin is sufficient to transform a NE state to aggressive NEUROD1+ SCLC. This finding provides a solid rationale for using the human ESC system to better understand the function of human oncogenes and tumor suppressors from a neuroendocrine origin.Weaknesses:(1) There is a major concern about the conclusion that MYC "yields a larger neuroendocrine compartment" related to Figures 4C and 4G, which is inadequately supported and likely inaccurate. There is overwhelming published data that while MYC can promote NEUROD1, it also tends to correlate with reduced ASCL1 and reduced NE fate (Mollaoglu et al, Cancer Cell, 2017; Zhang et al, TLCR, 2018; Ireland et al, Cancer Cell, 2020; Patel et al, Sci Advances, 2021). Most importantly, there is a lack of in vivo RP tumor controls to make the proper comparison to judge MYC's impact on neuroendocrine identity. RPM tumors are largely neuroendocrine compared to in vitro conditions, but since RP control tumors (in vivo) are missing, it is impossible to determine whether MYC promotes more or less neuroendocrine fate than RP controls. It is not appropriate to compare RPM tumors to in vitro RP cells when it comes to cell fate. Upon inspection of the sample identity in S1B, the fibroblast and basal-like cells appear to only grow in vitro and are not well represented in vivo; it is, therefore, unclear whether these are transformed or even lack RB/P53 or express MYC. Indeed, a close inspection of Figure S1B shows that RPM tumor cells have little ASCL1 expression, consistent with lower NE fate than expected in control RP tumors.

We would like to clarify two points related to the conclusions that we draw about MYC’s ability to promote an increase in the neuroendocrine fraction in hESC-derived cultures: (1) The comparisons in Figures 4C were made between cells isolated in culture following the standard 50 day differentiation protocol, where, following generation of LPs around day 25, MYC was added to the other factors previously shown to enrich for a PNEC phenotype (shRB1, shTP53, and DAPT). Therefore, the argument that MYC increased the frequency of “neuroendocrine cells” (which we define by a gene expression signature) is a reasonable conclusion in the system we are using; and (2) following injection of these cells into immunocompromised mice, an ASCL1-low / NEUROD1-high presentation is noted (Supplemental Figures 1F-G). In the few metastases that we were able use to sequence bulk RNA, there is an even more pronounced increase in expression of NEUROD1 with a decrease in ASCL1.

Some confusion may have arisen from our previous characterization of neuroendocrine (NE) cells using either ASCL1 or NEUROD1 as markers. To clarify, we have now designated cells positive for ASCL1 as classical NE cells and those positive for NEUROD1 as the NE variant. According to this revised classification, our findings indicate that MYC expression leads to an increase in the NEUROD1+ NE variant and a decrease in ASCL1+ classical NE cells. This adjustment has been reflected on the results section titled, “Inoculation of the renal capsule facilitates metastasis of the RUES2-derived RPM tumors” of the manuscript.

From the limited samples in hand, we compared the expression of ASCL1 and NEUROD1 in the weakly tumorigenic hESC RP cells after successful primary engraftment into immunocompromised mice. As expected, the RP tumors were distinguished by the lack of expression of NEUROD1, compared to levels observed in the RPM tumors.

In addition, since MYC appears to require Notch signaling to induce NE fate (cf Ireland et al), the presence of DAPT in culture could enrich for NE fate despite MYC's presence. It's important to clarify in the legend of Fig 4A which samples are used in the scRNA-seq data and whether they were derived from in vitro or in vivo conditions (as such, Supplementary Figure S1B should be provided in the main figure). Given their conclusion is confusing and challenges robustly supported data in other models, it is critical to resolve this issue properly. I suspect when properly resolved, MYC actually consistently does reduce NE fate compared to RP controls, even though tumors are still relatively NE compared to completely distinct cellular identities such as fibroblasts.

We have clarified the source of tumor sequencing data and the platform (single cell or bulk) in Figure 4 and Supplemental Figure 1. To reiterate – the RNA sequencing results from paired metastatic and primary tumors from the RPM model are derived from bulk RNA; the single cell RNA data in RP or RPM datasets are from cells in culture. These distinctions are clarified in the legend to Supplemental Figure 1.

(2) The rigor of the conclusions in Figure 1 would be strengthened by comparing an equivalent number of RP animals in the renal capsule assay, which is n = 6 compared to n = 11-14 in the MYC conditions.

As we did not perform a power calculation to determine a sample size required to draw a level of statistical significance from our conclusions, this comment is not entirely accurate. Our statistical rigor was limited by the availability of samples from the RP tumor model.

(3) Statistical analysis is not provided for Figures 2A-2B, and while the results are compelling, may be strengthened by additional samples due to the variability observed.

We acknowledge that the cohorts are relatively small but we have added statistical comparisons in Figure 2B.

(4a) Related to Figure 3, primary tumors and liver metastases from RPM or RPM-T58A-expressing cells express NEUROD1 by immunohistochemistry (IHC) but the putative negative controls (RP) are not shown, and there is no assessment of variability from tumor to tumor, ie, this is not quantified across multiple animals.

The results of H&E and IF staining for ASCL1, NEUROD1, CGRP, and CD56 in negative control (RP tumors) are presented in the updated Figure 3F-G.

(4b) Relatedly, MYC has been shown to be able to push cells beyond NEUROD1 to a double-negative or YAP1+ state (Mollaoglu et al, Cancer Cell, 2017; Ireland et al, Cancer Cell, 2020), but the authors do not assess subtype markers by IHC. They do show subtype markers by mRNA levels in Fig 4B, and since there is expression of ASCL1, and potentially expression of YAP1 and POU2F3, it would be valuable to examine the protein levels by IHC in control RP vs. RPM samples.

YAP1 positive SCLC is still somewhat controversial, so it is not clear what value staining for YAP1 offers beyond showing the well-established markers, ASCL1 and NEUROD1.

(5) Given that MYC has been shown to function distinctly from MYCL in SCLC models, it would have raised the impact and value of the study if MYC was compared to MYCL or MYCL fusions in this context since generally, SCLC expresses a MYC family member. However, it is quite possible that the control RP cells do express MYCL, and as such, it would be useful to show.

We now include Supplemental Figure S2 to illustrate four important points raised by this reviewer and others: (1) expression of MYC family members in the merged dataset (RP and RPM) is low or undetectable in the basal/fibroblast cultures; (2) MYC does have a weak correlation with EGFP in the neuroendocrine cluster when either WT MYC or T58A MYC is overexpressed; (3) MYCL and MYCN are detectable, but at low levels compared to CMYC; and (4) Expression of ASCL1 is anticorrelated with MYC expression across the merged single cell datasets using RP and RPM models.

**Reviewer #3 (Public Review):**
Summary:The authors continue their study of the experimental model of small cell lung cancer (SCLC) they created from human embryonic stem cells (hESCs) using a protocol for differentiating the hESCs into pulmonary lineages followed by NOTCH signaling inactivation with DAPT, and then knockdown of TP53 and RB1 (RP models) with DOX inducible shRNAs. To this published model, they now add DOX-controlled activation of expression of a MYC or T58A MYC transgenes (RPM and RPMT58A models) and study the impact of this on xenograft tumor growth and metastases. Their major findings are that the addition of MYC increased dramatically subcutaneous tumor growth and also the growth of tumors implanted into the renal capsule. In addition, they only found liver and occasional lung metastases with renal capsule implantation. Molecular studies including scRNAseq showed that tumor lines with MYC or T58A MYC led surprisingly to more neuroendocrine differentiation, and (not surprisingly) that MYC expression was most highly correlated with NEUROD1 expression. Of interest, many of the hESCs with RPM/RPMT58A expressed ASCL1. Of note, even in the renal capsule RPM/RPMT58A models only 6/12 and 4/9 mice developed metastases (mainly liver with one lung metastasis) and a few mice of each type did not even develop a renal sub capsule tumor. The authors start their Discussion by concluding: " In this report, we show that the addition of an efficiently expressed transgene encoding normal or mutant human cMYC can convert weakly tumorigenic human PNEC cells, derived from a human ESC line and depleted of tumor suppressors RB1 and TP53, into highly malignant, metastatic SCLC-like cancers after implantation into the renal capsule of immunodeficient mice.".Strengths:The in vivo study of a human preclinical model of SCLC demonstrates the important role of c-Myc in the development of a malignant phenotype and metastases. Also the role of c-Myc in selecting for expression of NEUROD1 lineage oncogene expression.Weaknesses:There are no data on results from an orthotopic (pulmonary) implantation on generation of metastases; no comparative study of other myc family members (MYCL, MYCN); no indication of analyses of other common metastatic sites found in SCLC (e.g. brain, adrenal gland, lymph nodes, bone marrow); no studies of response to standard platin-etoposide doublet chemotherapy; no data on the status of NEUROD1 and ASCL1 expression in the individual metastatic lesions they identified.

We have acknowledged from the outset that our study has significant limitations, as noted by this reviewer, and we explained in our initial letter of response why we need to present this limited, but still consequential, story at this time.

In particular, while we have attempted orthotopic transplantations of RPM tumor cells into NSG mice (by tail vein or intra-pulmonary injection, or intra-tracheal instillation of tumor cells), these methods were not successful in colonizing the lung. Additionally, we have compared the efficacy of platinum/etoposide to that of removing DOX in established RPM subcutaneous tumors, but we chose not to include these data as we lacked a chemotherapy responsive tumor model, and thus could not say with confidence that the chemotherapeutic agants were active and that the RPM models were truly resistant to standard SCLC chemotherapy. In a discussion about other metastatic sites, we have now included the following text:

“In animals administered DOX, histological examinations showed that approximately half developed metastases in distant organs, including the liver or lung (Figure 1D). No metastases were observed in the bone, brain, or lymph nodes. For a more detailed assessment, future studies could employ more sensitive imaging methods, such as luciferase imaging.”

**Recommendations for the authors:**

**Reviewer #2 (Recommendations For The Authors):**
Technical points related to Major Weakness #1:For Figure 4: Cells were enriched for EGFP-high cells only, under the hypothesis that cells with lower EGFP may have silenced expression of the integrated vector. Since EGFP is expressed only in the shRB1 construct, selection for high EGFP may inadvertently alter/exclude heterogeneity within the transformed population for the other transgenes (shP53, shMYC/MYC-T58A). Can authors include data to show the expression of MYC/MYC T58A in EGFP-high v -med v-low cells? MYC levels may alter the NEdifferentiation status of tumor cells.

Please now refer to Supplemental Figure S2.

Related to the appropriateness of the methods for Figure 4C, the authors state, "We performed differential cluster abundance analysis after accounting for the fraction of cells that were EGFP+". If only EGFP+ cells were accounted for in the analysis for 4C, the majority of RP cells in the "Neuroendocrine differentiated" cluster would not be included in the analysis (according to EGFP expression in Fig S1A-B), and therefore inappropriately reduce NE identity compared to RPM samples that have higher levels of EGFP.There is no consideration or analysis of cell cycling/proliferation until after the conclusion is stated. Yet, increased proliferation of MYC-high vs MYC-low cultures would enhance selection for more tumors (termed "NE-diff") than non-tumors (basal/fibroblast) in 2D cultures.The expression of MYC itself isn't assessed for this analysis but assumed, and whether higher levels of MYC/MYC-T58A may be present in EGFP+ tumor cells that are in the NE-low populations isn't clear. Can MYC-T58A/HA also be included in the reference genome?

We did not include an HA tag in our reference transcriptome. For [some] answers to this and other related questions, please refer to Supplemental Figure S2.

**Reviewer #3 (Recommendations For The Authors):**
(1) The experiments are all technically well done and clearly presented and represent a logical extension exploring the role of c-Myc in the hESC experimental model system.

We appreciate this supportive comment!

(2) It is of great interest that both the initial RP model only forms "benign" tumors and that with the addition of a strong oncogene like c-myc, where expression is known to be associated with a very bad prognosis in SCLC, that while one gets tumor formation there are still occasional mice both for subcutaneous and renal capsule test sites that don't get tumors even with the injection of 500,000 RPM/RPMT58A cells. In addition, of the mice that do form tumors, only ~50% exhibit metastases from the renal sub-capsule site. The authors need to comment on this further in their Discussion. To me, this illustrates both how incredibly resistant/difficult it is to form metastases, thus indicating the need for other pathways to be activated to achieve such spread, and also represents an opportunity for further functional genomic tests using their preclinical model to systematically attack this problem. Obvious candidate genes are those recently identified in genetically engineered mouse models (GEMMs) related to neuronal behavior. In addition, we already know that full-fledged patient-derived SCLC when injected subcutaneously into immune-deprived mice don't exhibit metastases - thus, while the hESC RPM result is not surprising, it indicates to me the power of their model (logs less complicated genetically than a patient SCLC) to sort through a mechanism that would allow metastases to develop from subcutaneous sites. The authors can point these things out in their Discussion section to provide a "roadmap" for future research.

Although we remain mindful of the relatively small cohorts we have studied, the thrust of Reviewer #3’s comments is now included in the Discussion. And there is, of course, a lot more to do, and it has taken several years already to get to this point. Additional information about the prolonged gestation of this project and about the difficulties of doing more in the near future was described in our initial response to reviewers/Editor, included near the start of this letter.

(3) I will state the obvious that this paper would be much more valuable if they had compared and contrasted at least one of the myc family members (MYCL or MYCN) with the CMYC findings whatever the results would be. Most SCLC patients develop metastases, and most of their tumors don't express high levels of CMYC (and often use MYCL). In any event, as the authors Discuss, this will be an important next stage to test.

We have acknowledged and explained the limitations of the work in several ways. Further, we were unaware of the relationship between metastases and the expression of MYC and MYCL1 noted by the reviewer; we will look for confirmation of this association in any future studies, although we have not encountered it in current literature.

(4) Their assays for metastases involved looking for anatomically "gross" lesions. While that is fine, particularly given that the "gross" lesions they show in figures are actually pretty small, we still need to know if they performed straightforward autopsies on mice and looked for other well-known sites of metastases in SCLC patients besides liver and lung - namely lymph nodes, adrenal, bone marrow, and brain. I would guess these would probably not show metastatic growth but with the current report, we don't know if these were looked for or not. Again, while this could be a "negative" result, the paper's value would be increased by these simple data. Let's assume no metastases are seen, then the authors could further strengthen the case for the value of their hESC model in systematically exploring with functional genomics the requirements to achieve metastases to these other sites.

We have included descriptions of what we found and didn’t find at other potential sites of metastasis in the results section, with the following sentences:

“In animals administered DOX, histological examinations showed that approximately half developed metastases in distant organs, including the liver or lung (Figure 1D). No metastases were observed in the bone, brain, or lymph nodes. For a more detailed assessment, future studies could employ more sensitive imaging methods, such as luciferase imaging.”

(5) Related to this, we have no idea if the mice that developed liver metastases (or the one mouse with lung metastasis) had more than one metastatic site. They will know this and should report it. Again, my guess is that these were isolated metastases in each mouse. Again, they can indicate the value of their model in searching for programs that would increase the number of the various organs.

We appreciate the suggestion. We observed that one of the mice developed metastatic tumors in both the liver and lungs. This information has been incorporated into the Results section.

(6) While renal capsule implantation for testing growth and metastatic behavior is reasonable and based on substantial literature using this site for implantation of patient tumor specimens, what would have increased the value of the paper is knowing the results from orthotopic (lung implantation). Whatever the results were (they occurred or did not occur) they will be important to know. I understand the "future experiments" argument, but in reading the manuscript this jumped out at me as an obvious thing for the authors to try.

We conducted orthotopic implantation several ways, including via intra-tracheal instillation of 0.5 million RP or RPM cells in PBS per mouse. However, none of the subjects (0/5 mice) developed tumor-like growths and the number of animals used was small. Further, this outcome could be attributed to biological or physical factors. For instance, the conducting airway is coated with secretory cells producing protective mucins and may not have retained the 0.5 million cells. This is one example that may have hindered effective colonization. Future adjustments, such as increasing the number of cells, embedding them in Matrigel, or damaging the airway to denude secretory cells and trigger regeneration might alter the outcomes. These ideas might guide future work to strengthen the utility of the models.

(7) Another obvious piece of data that would have improved the value of this manuscript would be to know whether the RPM tumors responded to platin-etoposide chemotherapy. Such data was not presented in their first RP hESC notch inhibition paper (which we now know generated what the authors call "benign" tumors). While I realize chemotherapy responses represent other types of experiments, as the authors point out one of the main reasons they developed their new human model was for therapy testing. Two papers in and we are all still asking - does their model respond or not respond dramatically to platin-etoposide therapy? Whatever the results are they are a vital next step in considering the use of their model.

Please see the comments above regarding our decision not to include data from a clinical trial that lacked appropriate controls.

(8) The finding of RPM cells that expressed NEUROD1, ASCL1, or both was interesting. From the way the data were presented, I don't have a clear idea which of these lineage oncogenes the metastatic lesions from ~11 different mice expressed. Whatever the result is it would be useful to know - all NEUROD1, some ASCL1, some mixed etc.

Based on the bulk RNA-sequencing of a few metastatic sites (Figure 4H), what we can demonstrate is that all sites were NEUROD1 and expressed low or no detectable ASCL1.

(9) While several H&E histologic images were presented, even when I enlarged them to 400% I couldn't clearly see most of them. For future reference, I think it would be important to have several high-quality images of the RP, RPM, RPMT58A subcutaneous tumors, sub-renal capsule tumors, and liver and lung metastatic lesions. If there is heterogeneity in the primary tumors or the metastases it would be important to show this. The quality of the images they have in the pdf file is suboptimal. If they have already provided higher-quality images - great. If not, I think in the long run as people come back to this paper, it will help both the field and the authors to have really great images of their tumors and metastases.

We have attempted to improve the quality of the embedded images. Digital resolution is a tradeoff with data size – higher resolution images are always available upon request, but may not be suitable for generation of figures in a manuscript viewed on-line.